# LipidII interaction with specific residues of *Mycobacterium tuberculosis* PknB extracytoplasmic domain governs its optimal activation

Prabhjot Kaur[1], Marvin Rausch [2,3], Basanti Malakar[1], Uchenna Watson[4], Nikhil P. Damle[1,6], Yogesh Chawla[1,7], Sandhya Srinivasan[5], Kanika Sharma[5], Tanja Schneider[2,3], Gagan Deep Jhingan[5], Deepak Saini[4], Debasisa Mohanty[1], Fabian Grein [2,3] & Vinay Kumar Nandicoori [1]

The *Mycobacterium tuberculosis* kinase PknB is essential for growth and survival of the pathogen in vitro and in vivo. Here we report the results of our efforts to elucidate the mechanism of regulation of PknB activity. The specific residues in the PknB extracytoplasmic domain that are essential for ligand interaction and survival of the bacterium are identified. The extracytoplasmic domain interacts with mDAP-containing LipidII, and this is abolished upon mutation of the ligand-interacting residues. Abrogation of ligand-binding or sequestration of the ligand leads to aberrant localization of PknB. Contrary to the prevailing hypothesis, abrogation of ligand-binding is linked to activation loop hyperphosphorylation, and indiscriminate hyperphosphorylation of PknB substrates as well as other proteins, ultimately causing loss of homeostasis and cell death. We propose that the ligand-kinase interaction directs the appropriate localization of the kinase, coupled to stringently controlled activation of PknB, and consequently the downstream processes thereof.

[1] National Institute of Immunology, Aruna Asaf Ali Marg, New Delhi 110067, India. [2] Institute for Pharmaceutical Microbiology, University Hospital Bonn, University of Bonn, Bonn 53105, Germany. [3] German Center for Infection Research (DZIF), Partner Site Bonn-Cologne, Bonn 53105, Germany. [4] Department of Molecular Reproduction, Development and Genetics, Indian Institute of Science, Bengaluru 560012, India. [5] Vproteomics, Valerian Chem Private Limited, Green Park Main, New Delhi 110016, India. [6] Present address: BIOSS, Center for Biological Signaling Studies, University of Freiburg, Freiburg 79104, Germany. [7] Present address: Department of Microbiology and Immunology, Weill Cornell Medical College, New York 10065 NY, USA. These authors contributed equally: Marvin Rausch, Basanti Malakar. Correspondence and requests for materials should be addressed to V.K.N. (email: vinaykn@nii.ac.in)

P rotein phosphorylation has come forth as a preeminent circuitry regulating a vast number of physiological processes in the bacterial kingdom. A particular class of receptor-type serine-threonine kinase called PASTA (Penicillin binding proteins And Serine Threonine Associated) kinase is widespread across gram-positive firmicutes and actinomycetes and is known for its functions associated with bacterial cell growth[1]. These protein kinases have an intracellular kinase domain, which shows sequence and structural homology to the eukaryotic serine/threonine kinases, and an extracytoplasmic (Ec) domain made up of varying number of PASTA domains. PASTA kinases are usually required by bacteria under stress conditions like nutrient starvation, antibiotic stress, biofilm formation etc., and are non-essential for their vegetative growth[2]. However, in the pathogenic bacterium *Mycobacterium tuberculosis* (*Mtb*) the PASTA kinase PknB (PknB$_{Mtb}$) is an essential gene[3–5] and is proposed to be one of the master regulators of serine/threonine phosphorylation-mediated signaling[6].

The essential nature of PknB in mycobacteria stems from its ability to influence the activity of a large repertoire of substrates involved in cell wall synthesis, cell growth, cellular metabolism, transcription, and translation[7]. Over-expression or depletion of PknB impacts cellular morphology and survival of *Mtb*[5,8], which suggests that the expression and activity of this kinase must be critically fine-tuned inside the bacterium. PknB levels are modulated under different conditions of mycobacterial growth: for instance its levels are down regulated during dormancy[9] and nutrient starvation[10] and are up regulated during exponential growth[8] and resuscitation[9]. The dynamicity of PknB$_{Mtb}$ regulation implies that the receptor kinase actively monitors its environment and responds accordingly, in an effort to provide survival advantage. PknB is believed to respond to environmental signals through PASTA domain interactions with the specific ligand, identified to be non-crosslinked peptidoglycan (PG) fragments called muropeptides[11]. In line with this, purified PASTA domains of PknB$_{Mtb}$ interact and bind with a synthetic muropeptide containing isoglutamine (iGln) and meso-diaminopimelic acid (mDAP) residues at the second and third position of the stem peptide in vitro (Fig. 1a)[11,12].

The prevailing hypothesis suggests that the interaction of the extracytoplasmic domain with the ligand results in the dimerization of intracellular kinase domain, which is required for the activation of the kinase through activation loop phosphorylations (Fig. 1b)[13]. The hypothesis is based on the front-to-front and back-to-back dimeric crystal structures of cytosolic kinase domain[14,15] and surface plasmon resonance-based in vitro binding experiments of PASTA domain with the muropeptides[11]. In consonance with this we have previously reported that the extracytoplasmic PASTA domains are indispensable for the function of PknB and survival of *Mtb*[5]. Deletion of the terminal PASTA domain (PASTA4) alone results in compromised survival[5], suggesting that it plays a leading role in kinase-ligand interactions. To date, the hypothesis with respect to PknB$_{Mtb}$ activation has not been tested in vivo. Here we set out to answer the following questions: (i) Is PASTA4 sufficient for PknB$_{Mtb}$ signaling?, (ii) What are the ligand binding residues in the extracytoplasmic domain?, (iii) What is the impact of abrogating ligand binding on the localization and activation of PknB?, (iv) What are the physiological ligands that interact with PASTA domains? and (v) What is the impact of abrogating ligand binding on the phosphorylation of target substrates of PknB?

Here we identify ligand interacting residues and show that mutating these residues caused abolition of ligand binding. Abrogation of ligand binding triggers aberrant localization and hyperactivation of PknB, which in turn results in hyperphosphorylation of both canonical and non-canonical downstream

target substrates, eventually leading to cell death. Results suggest that interaction with the ligand is critical for appropriate localization and regulation of the kinase activity.

## Results

### PASTA4 and domain length are essential for PknB function.
We have previously shown that the deletion of the PASTA4 domain (Fig. 1c; PknB-123) compromises the in vivo functionality of PknB[5]. The terminal PASTA domain of StkP, a *Streptococcus pneumoniae* ortholog of the PknB$_{Mtb}$, was demonstrated to be necessary and sufficient for it's signaling[16]. Thus we sought to determine the role of the terminal PASTA4 in the context of shorter total domain length. We employed previously described *Mtb* conditional mutant of *pknB* (*RvΔB*), wherein the native locus has been modified to bring its expression under pristinamycin inducible promoter, which allows the bacterium to grow efficiently in the presence of pristinamycin but not in its absence[5,17] (Supplementary Fig. 1a). To assess the effect of PknB mutations, we generated wild type or mutant constructs in pNit-3F (Supplementary Fig. 1b), which could be induced with isovaleronitrile and owing to the presence of 3X-FLAG tag, the ectopically expressed PknB migrates slower compared with the endogenous protein (Supplementary Fig. 1c).

To investigate the role of PASTA4 as well as the domain length, we have generated PknB-234, lacking PASTA1 wherein the terminal PASTA4 is retained, and PASTA-1212, a chimera of appropriate domain length where PASTA1-2 were repeated (Fig. 1c). Western blot analysis confirmed the efficient depletion of PknB in the absence of pristinamycin as well as efficient expression of 3X-FLAG tagged wild-type and mutant proteins (Fig. 1d). On examining the ability of wild-type and mutant PknB proteins to complement the in vivo functionality of PknB (Fig. 1e), it was observed that vector-transformed *RvΔB* showed significantly compromised survival in the absence of pristinamycin while the ectopic expression of 3F-PknB rescued the growth defects (Fig. 1e). Even though PknB-234 was marginally better compared to either PknB-123 or PknB-1212 (~1.5 vs 2 log fold) in rescuing the growth defects, growth was significantly compromised in all three strains when compared with the wild type (Fig. 1e). To assess the impact in an infection scenario, we evaluated the survival in differentiated THP1 cells (Fig. 1f). The data resembled the in vitro growth results, wherein PknB-234 showed slightly better survival as compared with PknB-1212 or PknB-123. These results suggest that while the PASTA4 domain is quite distinct and important, yet the appropriate length of the entire domain is also vital for efficient PknB$_{Mtb}$ function.

### mDAP and iGln interacting residues influence the survival.
The structure of the PASTA domains has been derived with the help of NMR as well as X-ray crystallography[18,19]. Nonetheless, the residues that are responsible for the interaction with the proposed muropeptide ligand have not yet been identified. As PASTA4 is critical for PknB functionality, we examined a possible role for it in ligand binding. As PknB is hypothesized to dimerize upon ligand binding[14], we assumed that the PASTA domains interact with the muropeptide as a dimer. With the help of in silico molecular simulations, we identified a potential dimerization interface between the PASTA4 domains (Fig. 2a), which predominantly comprises of residues that are conserved across the PASTA4 domains of prokaryotic kinases (Supplementary Fig. 2). The initial simulation data suggested that ligand binding might be extending into the PASTA3 domain; hence, the dimeric PASTA3-4 domain was employed for further analysis. The ligand-binding domain majorly comprised of residues in the linker regions between PASTA3 and 4 domains (Fig. 2a &

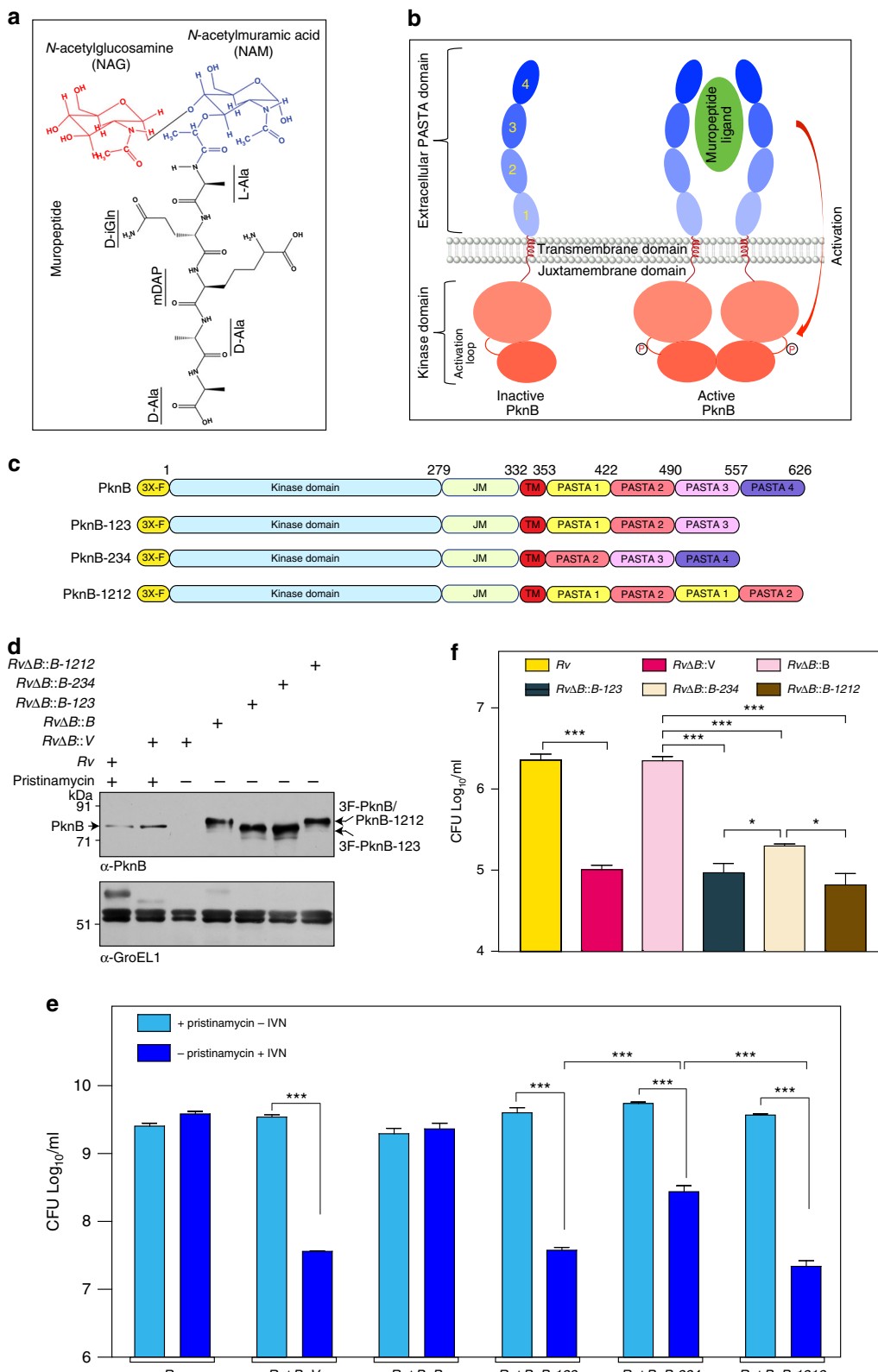

Supplementary Fig. 2). Data suggested that $Ser_{556}$ and $Lys_{557}$ (SK) interact with the carboxy terminal region of the iGln and $Asn_{559}$ and $Gln_{560}$ (NQ) interact with the mDAP residues in the muropeptide ligand (Fig. 2a & Supplementary Fig. 2a). We set out to assess the impact of mutating these putative iGln and mDAP binding residues in modulating PknB functionality. We generated a PknB tetra mutant (PknB-GM), wherein all four putative ligand

interacting residues have been mutated simultaneously (Fig. 2b). The $Rv\Delta B$ strain was transformed with pNit-3F-PknB and pNit-3F-PknB-GM and the expression of both proteins was confirmed (Fig. 2c). While the wild type could successfully rescue the phenotype both in vitro and ex vivo, the tetra mutant failed to do so (Fig. 2d, e). The growth phenotype did not vary even when the expression of PknB from pNit-3F constructs was not induced

**Fig. 1** PASTA4 and domain length are essential for PknB function. **a** Chemical structure of suggested muropeptide ligand for PknB. **b** Schematic outline of prevailing hypothesis vis-a-vis PknB activation, wherein the ligand binding is proposed to result in dimerization and phosphorylation of the activation loop residues in the intracellular kinase domain. **c** Schematic representation of full-length PknB, PknB-PASTA domain deletion mutants, and the chimera. **d** $Rv\Delta B$ strain was electroporated with pNit-3F, pNit-3F-PknB, pNit-3F-PknB-123, pNit-3F-PknB-234, and pNit-3F-PknB-1212 to generate $Rv\Delta B::V$, $Rv\Delta B::B$, $Rv\Delta B::123$, $Rv\Delta B::234$, and $Rv\Delta B::1212$. Whole-cell lystaes (WCLs) were prepared from cultures initiated at $A_{600}$ of ~0.05 and grown for 5 days in the presence or absence of of pristinamycin. IVN was added in the absence of pristinamycin to the cultures to induce the expression of episomal copy. Ten microgram each of WCLs were resolved on 8% SDS-PAGE, transferred to nitrocellulose membrane, and western blotted with α-PknB and α-GroEL1 antibodies. The endogenous and episomally expressed PknB are indicated by arrow. **e** $Rv$ and $Rv\Delta B$ transformants were inoculated at initial $A_{600}$ of ~0.05 and grown in the presence or absence of pristinamycin. IVN (0.2 μM) was added in the absence of pristinamycin to the cultures to induce the expression of episomal copy. After 6 days of in vitro growth, Colony forming Units (CFU) were enumerated by plating appropriate dilutions on 7H11 agar plates containing pristinamycin. Data is representative of three biologically independent experiments with each experiment being performed in triplicates. **f** $Rv$ and $Rv\Delta B$ transformants grown in the presence of pristinamycin till $A_{600}$ of ~0.6–1.0 were washed thrice with PBS to remove inducer. These cultures were used to infect differentiated THP-1 human macrophages at 1:10 M.O.I. IVN was added in the media to induce the expression of episomal copy and CFUs were evaluated 72 h post infection (p.i). Data are representative of one of the two biologically independent experiments and each experiment was performed in triplicates. Statistical analysis was performed with the unpaired $t$-test using Graphpad software. Data shown represents mean + standard deviation (SD). $^{**}p < 0.005$, $^{***}p < 0.0005$. Source data are provided as a Source Data file

with isovaleronitrile, suggesting that the results are not due to overexpression artifacts (Supplementary Fig. 2b, c). Thus, putative iGln and mDAP interacting residues in the PASTA domain of PknB seem to be necessary for its functionality.

**iGln/mDAP interacting residues are independently essential**. Next we sought to assess the impact of individually mutating the amino acid pairs, which we believed to interact with iGln or mDAP residues. Hence we generated PknB-G and PknB-M, wherein **SK** or **NQ** residues were mutated to **AA** or **DE** residues, respectively (Fig. 3a). Western blot showed efficient expression of 3F-PknB, 3F-PknB-G, and 3F-PknB-M in the complemented strains (Fig. 3b). Compared with the tetra mutant (Fig. 2d), strains complemented with either PknB-G or PknB-M showed partial growth defects, albeit to different extents (Fig. 3c, e). PknB-G ($Rv\Delta B::B$-G) showed one and half log-fold poorer survival compared with PknB ($Rv\Delta B::B$), while it showed ~3 log folds better survival compared with the control ($Rv\Delta B::V$) (Fig. 3c). On the other hand, PknB-M ($Rv\Delta B::B$-M) showed 3 log fold lower survival compared with $Rv\Delta B::B$ and one and half log-fold better survival compared with the control (Fig. 3e). These results suggested that mDAP interacting residues play a more critical role compared with iGln interacting residues. However in a THP1 infection model (Fig. 3d, f), both the mutants were equally abrogated and showed defects similar to the tetra mutant (Fig. 2e). THP1 infection experiment performed at lower MOI (1:4) showed similar defects as those observed at higher MOI (1:10) (Figs 2e and 3d, f). Data suggests that both iGln and mDAP interacting residues are individually critical for PknB functionality, especially during the ex vivo infection scenario where even marginal perturbations in ligand binding are not endured.

**Mutations in the PknB-Ec abrogate its binding to LipidII**. PG synthesis at the poles and septum region involves two stages. In the first stage, which happens in the cytoplasm, LipidII is synthesized from UDP-GlcNAc precursor by sequential action of multiple Mur family enzymes[20]. LipidII is composed of $N$-acetylglucosamine (NAG)-$N$-acetylmuramic acid (NAM)-pentapeptide (stem peptide) connected to the membrane anchored decaprenyl phosphate through a pyrophosphate link (Supplementary Fig. 3a). LipidII is then translocated/flipped into the periplasmic region, which provides the NAG-NAM-pentapeptide moieties to the growing PG layer[20]. The nature of the lipid moiety, the amino acids in the stem peptides and modification in the NAG and NAM sugars, vary from species to species[21]. The *Staphylococcus aureus* PknB ortholog ($PknB_{sa}$) has been shown to

interact strongly with LipidII through PASTA domains (1:2 molar ratio of LipidII:$PknB_{sa}$)[22]. Thus we sought to explore the possibility of an interaction between $PknB_{Mtb}$ and LipidII. Unlike *S. aureus*, wherein the LipidII stem peptide contains a lysine residue at the third position, the LipidII in *Mtb* carries mDAP at the analogous position in the stem peptide[21] (Supplementary Fig. 3a).

It is apparent from the data presented in Figs. 2 and 3 that PASTA linker mutants were defective in rescuing PknB function, which may be due to major conformational changes. To assess the conformation changes, His-tagged PknB and the mutants of extracytoplasmic PASTA domains (PknB-Ec) were purified and CD analysis was performed (Fig. 4a–c). CD analysis indicated absence of gross conformational changes in the secondary structure (Fig. 4a–c). Next we sought to examine the interaction between the PknB-Ec (Fig. 4d) and lysine- or mDAP-containing LipidII molecules (Supplementary Fig. 3b). When mDAP-containing LipidII was used in the assay, there was no detectable LipidII upon extraction at 4:1 molar ratio of PknB-Ec:LipidII, indicating a strong interaction (Fig. 4e). This interaction could be abrogated upon treating the complex with trypsin, which would degrade the protein (Fig. 4e). Interestingly, lysine-containing LipidII could not be sequestered into a complex by PknB-Ec even at a molar ratio of 8:1 (protein:LipidII; Fig. 4e). We then evaluated the ability of mutants to bind with mDAP-LipidII. It is evident from the data presented in Fig. 4f that neither double nor the tetra mutant showed any binding with the mDAP-LipidII (Fig. 4d, f). The results show that the extracytoplasmic domain of $PknB_{Mtb}$ binds specifically with mDAP-containing LipidII and the binding is abrogated upon mutating ligand-interacting residues.

**Abrogation of ligand binding perturbs localization of PknB**. Nascent PG biosynthesis takes place at the poles and mid cell (septum) regions[23] and hence the precursors such as LipidII and muropeptides are anticipated to be concentrated at these niches. Interestingly, PknB also localizes to both polar and mid-cell regions and the extracytoplasmic PASTA domains govern the localization[11]. Thus the rational supposition would be that the localization of PknB is probably dictated by the interaction between PASTA domain and PG precursors. However, the hypothesis has not been tested till date, perhaps due to the non-availability of PknB mutants that fail to interact with the ligand. We examined the localization of GFP-PknB and GFP-PknB-GM mutants in *M. smegmatis pknB* conditional mutant ($mc^2\Delta B$), following the reasoning that in the absence of endogenous PknB; GFP-PknB or GFP-PknB-GM would be the sole PknB in the cell

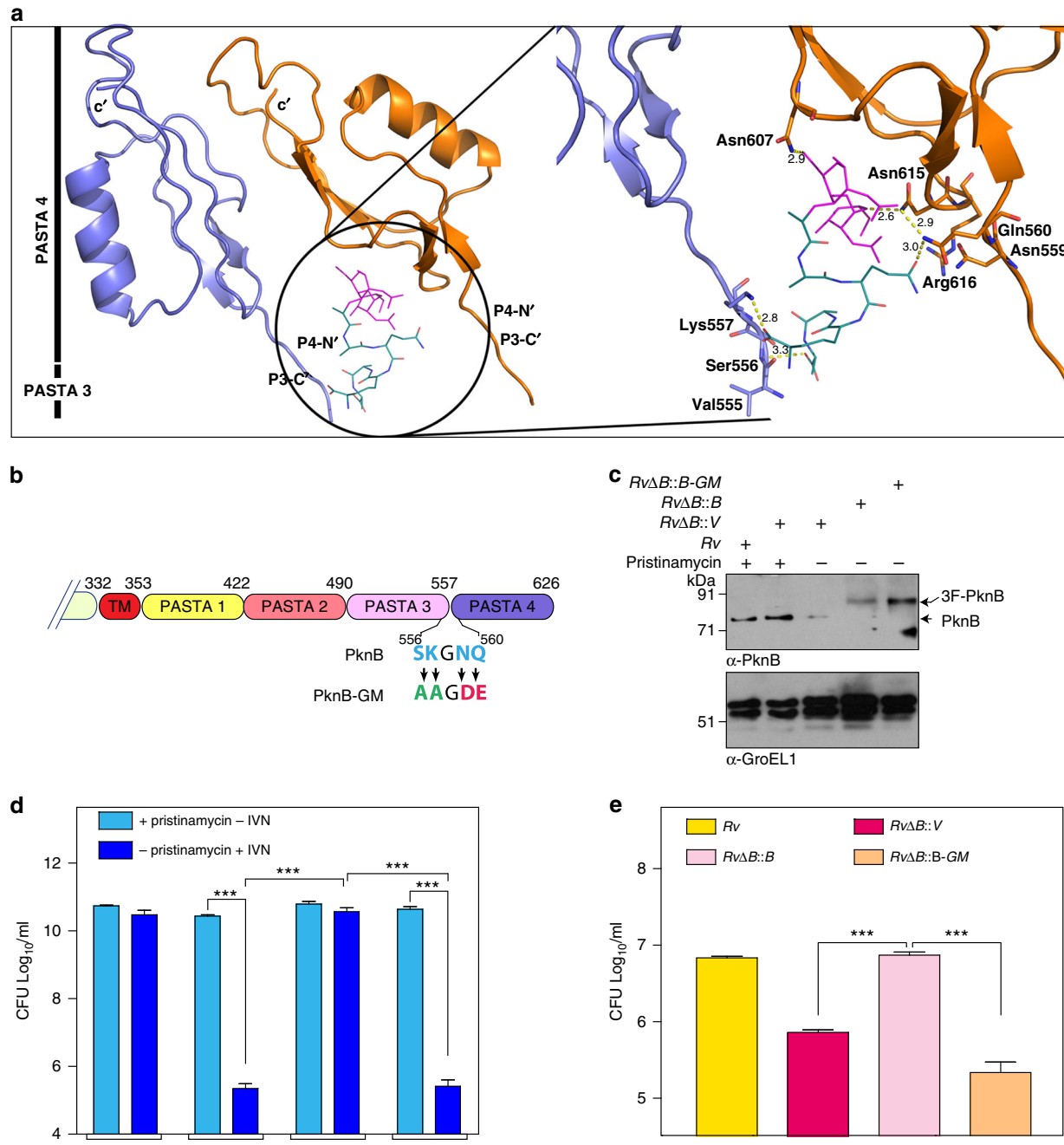

**Fig. 2** mDAP and iGln interacting residues influence the survival. **a** Model of *Mtb* PknB PASTA3-4 dimer in complex with the muropeptide ligand. Inset panel shows a magnified view of the interactions between the mDAP and iGln residues of the muropeptide with polar amino acids between the PASTA3 and 4 domains of PknB. **b** Schematic representation of extra-cytoplasmic PASTA domain of PknB and the residues in the PASTA3-4 linker region suggested to be involved in interaction with iGln (SK) and mDAP (NQ) residues in the muropeptide. The mutations introduced in the linker region to generate PknB-GM are indicated. **c** Western blot analysis of WCLs prepared from *Rv* and *RvΔB* transformants. The transformants were seeded at $A_{600}$ of ~0.05 in the presence of pristinamycin or IVN as indicated for 5 days and WCLs were resolved on 8% SDS-PAGE and probed with α-PknB and α-GroEL1 antibodies. **d** Cultures of *Rv* and *RvΔB* transformants were initiated at $A_{600}$ of ~0.05 and grown in the presence of 1.5 μg/ml pristinamycin or 0.2 μM IVN as indicated for 6 days. CFUs were evaluated by plating appropriate dilution on 7H11 agar plates containing pristinamycin. Data are representative of the three biologically independent experiments and each experiment was performed in triplicates. **e** Human monocytic cell line THP-1 was differentiated to macrophages and was infected at 1:10 M.O.I. with *Rv* and *RvΔB* transformants grown to $A_{600}$ of ~0.6–1.0 in the presence of pristinamycin. IVN was added in the media to induce the expression of episomal copy and CFUs were enumerated 72 h p.i. Data are representative of one of the two biological replicates and each experiment was performed in triplicates. Data represents mean + SD. Statistical analysis was performed with the unpaired *t*-test using Graphpad software. ***$p < 0.0005$. Source data are provided as a Source Data file

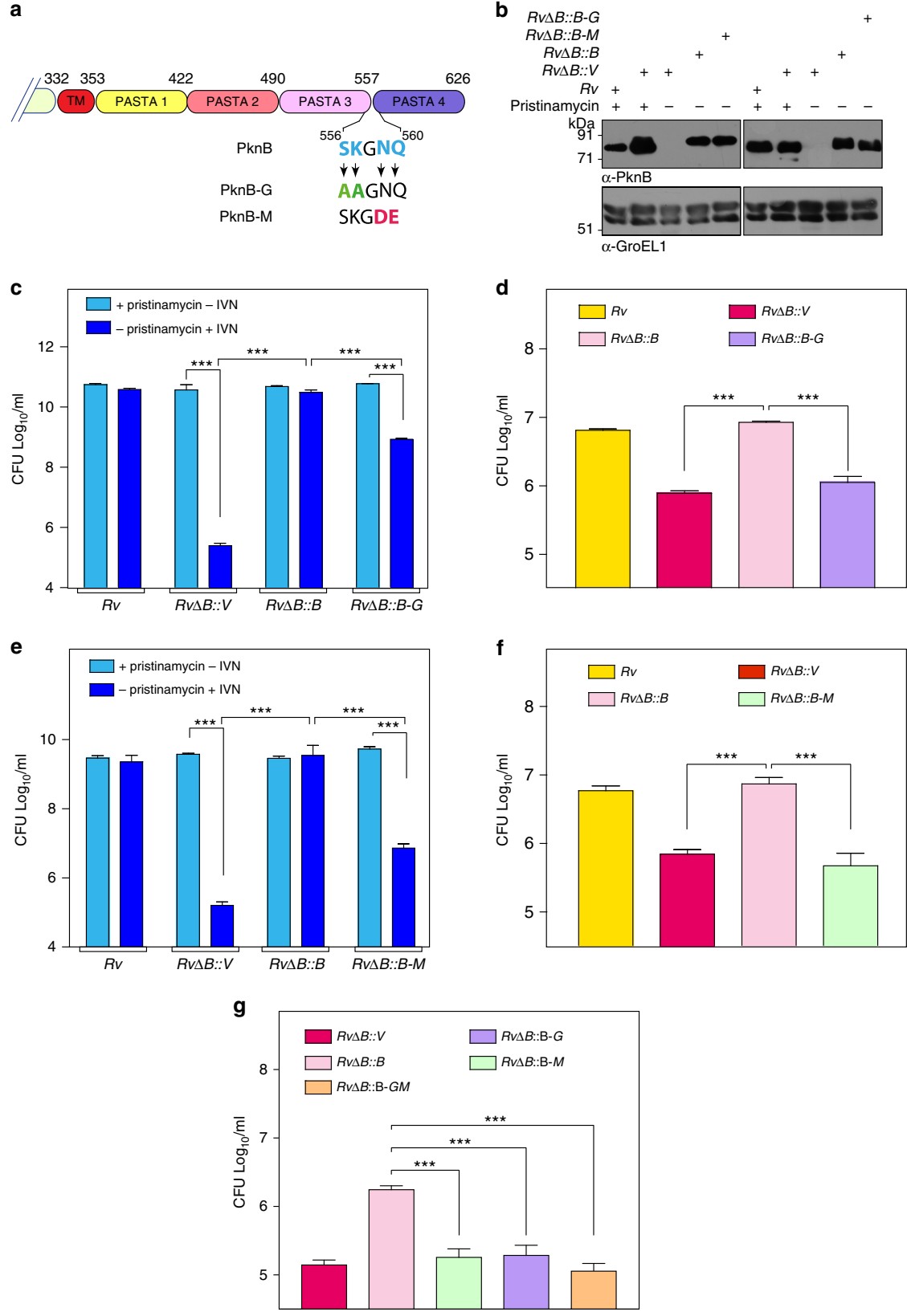

and hence phenotypic impact in terms of localization can be clearly visualized. Although GFP-PknB shows a strong punctate distribution at the polar and mid cell regions, GFP-PknB-GM mutant shows aberrant localization throughout the cell (Fig. 5a), suggesting that abrogation of ligand binding results in aberrant localization. Quantification of pole/septa vs. aberrant localization

drives home the message (Fig. 5b). If this was indeed accurate, the converse experiment wherein the ligand is sequestered should also result in aberrant localization. Thus, we asked what would happen to PknB localization when LipidII is sequestered into a complex with antibiotic "nisin", which binds through the pyrophosphate group resulting in membrane pore formation and

**Fig. 3** iGln or mDAP interacting residues are independently essential. **a** Schematic outline of the PknB and the PknB-mutants. The mutations introduced in the linker region to abrogate either iGln (SK → AA) or mDAP (NQ → DE) binding are indicated. **b** WCLs prepared from cultures of Rv and RvΔB transformants inoculated at $A_{600}$ of ~0.05 in the presence of pristinamycin or IVN and grown for 5 days were probed with α-PknB and α-GroEL1 antibodies. **c**, **e** Rv and RvΔB transformants were inoculated at initial $A_{600}$ of ~0.05 and grown for 6 days in the presence of pristinamycin or IVN. CFUs were enumerated on 7H9 plates containing pristinamycin. Data are representative of the three biologically independent experiments with each experiment being performed in triplicates. **d**, **f**, **g** Differentiated THP-1 cells were infected at 1:10 (**d**, **f**) or 1:4 (**g**) M.O.I. with Rv and RvΔB transformants grown to $A_{600}$ of ~0.6–1.0 in the presence of pristinamycin. Pristinamycin was removed from cultures prior to infection by washing with $PBST_{80}$ and PBS. IVN was added in the media to induce the expression of episomal copy and CFUs were enumerated 72 h p.i. Data are representative of one of the two biological experiments and each experiment was performed in triplicates. Data represents mean + SD. Statistical analysis was performed with the unpaired t-test using Graphpad software. ***$p < 0.0005$. Source data are provided as a Source Data file

eventual cell death[24–26]. The antibiotic isoniazid (INH) that does not interact with LipidII was used as the control (Fig. 5c). Although we observed distinct puncta with $mc^2ΔB::GFP-B$ in the absence of any antibiotic or in the presence of INH, aberrant number/mislocalized puncta were observed (>3) upon nisin treatment (Fig. 5c). Quantification showed a significant increase in the number of cells with more than 3 puncta or localization all through the cell (Fig. 5d). These results suggest that LipidII is likely to be the major intracellular ligand of PknB, and importantly, ligand binding is essential for the appropriate localization. Since the PknB-GM fails to interact with the ligand and does not localize to pole and septum, we reasoned that it might impact the diffusion dynamics. We investigated the rate of recovery of GFP-PknB and GFP-PknB-GM upon photobleaching by performing FRAP (Fluorescence Recovery After Photobleaching) experiments (Fig. 5e). Although GFP-PknB showed relatively slower recovery ($t_{1/2} = 111$ s), GFP-PknB-GM mutant showed substantially higher dynamicity with $t_{1/2} = 90.56$ s (Fig. 5f), suggesting that ligand binding also plays a role in modulating molecular dynamicity.

**PknB-GM is hyper-phosphorylated in the activation loop**. If the prevailing hypothesis with respect to the activation of PknB shown in Fig. 1b is accurate, abrogation of ligand binding should result in decreased phosphorylation of the activation loop residues. PknB is auto-phosphorylated at the T171 and T173 residues within the activation loop in vitro, and mutating these residues significantly diminishes its activity[27], (Fig. 6a). PknB is also phosphorylated in vitro in the juxtamembrane domain[27,28], (Fig. 6a); however, there are no reports about the influence of these phosphorylations its function. To scrutinize the roles of T171 and T173 residues, we generated PknB-TATA (T171A, T173A) mutant (Supplementary Figs. 4a and 6a), and the activity was analyzed using GarA as the substrate (Supplementary Fig. 4b–d). In agreement with the previous report[27], the PknB-TATA mutant showed significantly compromised activity (Supplementary Fig. 4c, d). To examine their in vivo role, PknB or PknB-TATA clones made using integrative vector as backbone were electroporated into RvΔB cells. The expression of PknB and PknB-TATA from the integrated construct was comparable to that of inducible PknB at the native locus (Fig. 6b). Compared with RvΔB::V, RvΔB::B-TATA showed better survival; nonetheless, the mutant was significantly compromised when compared with RvΔB::B, indicating the essentiality of activation loop phosphoryaltions (Fig. 6c). Importantly, in the ex vivo model of infection, both RvΔB::V and RvΔB::B-TATA were found to be equally compromised (Fig. 6d).

To determine the extent of activation loop and juxtamembrane phosphorylations, we resorted to isobaric TMT labeling (Fig. 6e). Western blot and quantitative mass spectrometry analysis of total proteome demonstrated depletion of PknB and similar expression of 3F-PknB and 3F-PknB-GM (Fig. 6f, g). However, the levels of PknB protein in RvΔB::B & RvΔB::B-GM

were ~1.2–1.5 ($log_2$) fold higher compared with PknB in RvΔB (+pristinamycin; Fig. 6g; Supplementary Table 1a). Intriguingly, contrary to the current belief, we observed hyperphosphorylation of both activation loop and juxtamembrane domain in the 3F-PknB-GM mutant compared with 3F-PknB (Fig. 6h & Supplementary Table 1b). Although the role of activation loop phosphorylation is known, the role of juxtamembrane phosphorylation needs further investigation. Thus the data suggests that abrogation of ligand binding results in hyper phosphorylation of PknB, strongly suggesting that the ligand binding plays a regulatory role.

**PknB-GM hyperphosphorylates specific & non-specific targets**. We identified a total of 632 (Supplementary Fig. 6a & Supplementary Data 1) and 258 (Supplementary Fig. 6b) proteins in the proteomic and phosphoproteomic samples, respectively. Of the 258 phosphoproteins, 242 had common phosphopeptides (total = 390 phosphopeptides) in all three replicates. Among them, 147 proteins encompassing 257 phosphopeptides were present in both, proteomic and phosphoproteomic samples (Supplementary Fig. 6c & Supplementary Data 2). The intensity ratios of these 257 phosphopeptides were normalized with respect to the whole-protein intensity ratios (Supplementary Data 1) and the mean normalized ratios (Supplementary Data 3) were converted to $log_2$ values (Supplementary Table 2) and plotted as a heatmap (Fig. 7a). The 147 phosphoproteins were distributed across all the functional categories (Fig. 7b) with a majority of the phosphorylations on threonine (68%), followed by serine (~23%) and tyrosine (~9%), residues, a trend universal to all mycobacterial phosphoproteomic studies[29–31]. Among the 257 phosphopeptides, a total of 111 phosphopeptides mapping to 73 different proteins (which included multiple well-characterized substrates such as Ef-Tu, Wag31, and HupB)[8,32,33] were potentially the products of PknB-mediated phosphorylation (Fig. 7b, c; Supplementary Table 3a). Recently, Carette et al. have identified 46 potential targets of PknA and PknB with the help of a specific inhibitor[34]. Nine of these identified targets were found to be among the 73 potential targets of PknB identified in this study (Supplementary Table 3b). Interestingly, we also identified 8 putative PknB-dependent tyrosine phosphorylated peptides (Fig. 7d; Supplementary Table 4a, b), in agreement with a previous study[35] where PknB was suggested to be a dual specificity kinase.

The analysis of the activation loop phosphorylations in PknB suggested that PknB-GM is hyperphosphorylated (Fig. 6h), which led us to infer it to be a hyperactive kinase. In line with this prediction, complementation with the PknB-GM mutant seems to have resulted in the remarkable hyperphosphorylation of cellular proteins (Fig. 7a; Supplementary Table 2). The data in Fig. 7a can be further divided into three clusters (Supplementary Table 4c): cluster 1 where the phosphorylation is unperturbed by depletion of PknB as well as complementation with ectopic expression of PknB or PknB-GM; cluster 2 representing direct

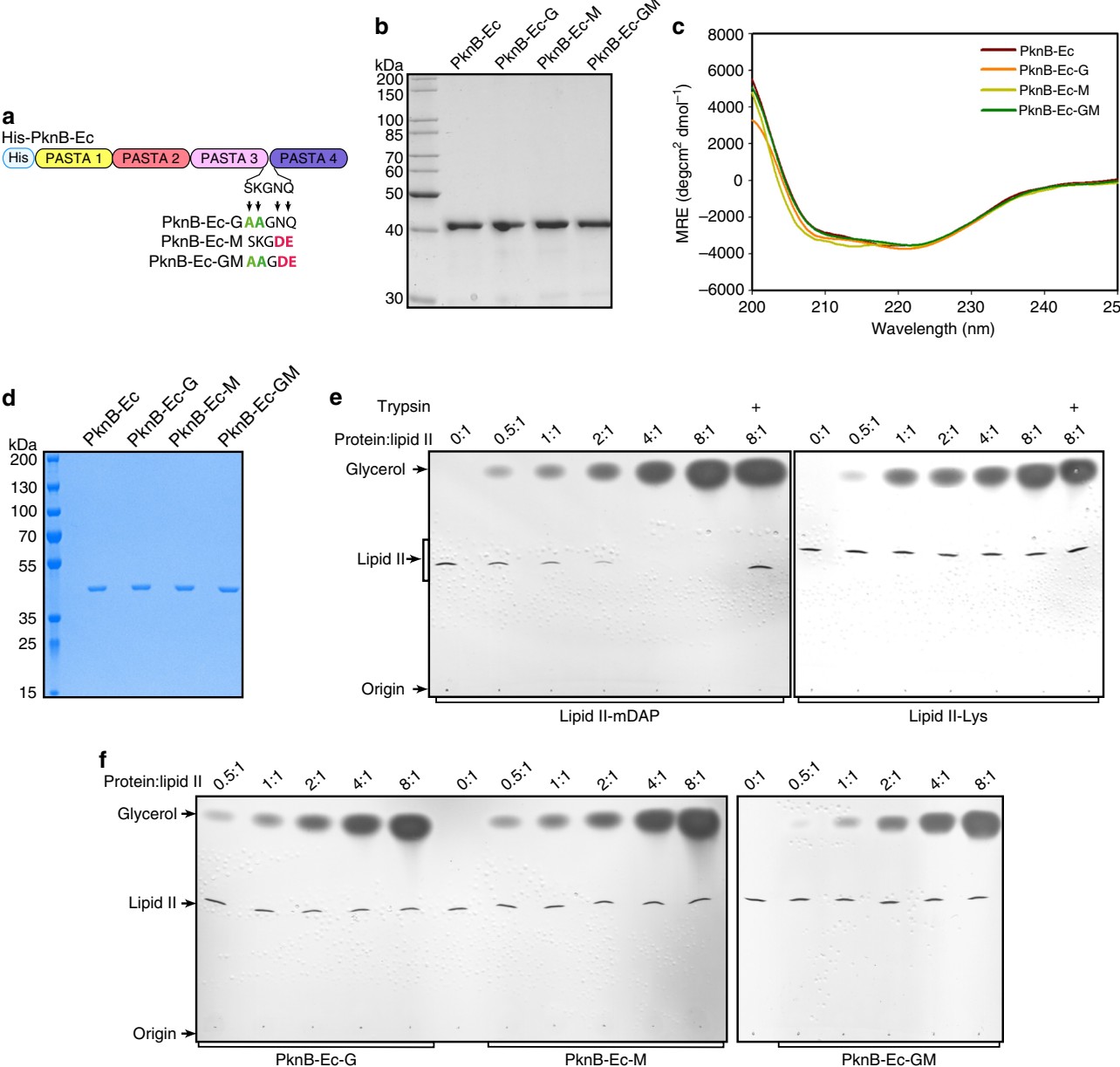

**Fig. 4** Abrogation of ligand binding perturbs localization of PknB. **a** Schematic depicting hexa-His tagged extracytoplasmic region of PknB (PknB-Ec) and PknB-Ec mutants. **b** His-PknB wild type and mutants were purified as described in methods and 2 μg of purified PknB-Ec$_{wt/mutant}$ proteins were resolved on SDS-PAGE and stained with commassie. The purified proteins were used for the CD experiment in **c**. **c** CD spectrum of PknB-EC$_{wt/mutant}$ proteins in far-UV range (200–250 nm) CD data is depicted as MRE values (deg cm$^2$ dmol$^{-1}$) in the Y-axis plotted against wavelength (nm) in X-axis. **d** 1 μg each of His-PknB-Ec$_{wt/mutant}$ purified for the LipidII binding experiment were resolved on SDS-PAGE and stained with coommassie as a loading control for experiments shown in **e**, **f**. **e** 2 nmol of mDAP or Lys containing LipidIIs were incubated with increasing mole:mole ratio of His-PknB-Ec$_{wt}$ with respect to LipidII. Addition of Trypsin to degrade His-PknB-Ec$_{wt}$ in the reaction mixture is indicated. The samples were extracted with BuOH/PyrAc and resolved on TLC to analyze the presence of extractable LipidII. LipidII trapped in a stable complex with PknB Ec$_{wt}$ resides in the water phase and free LipidII is extracted and migrates to a defined position on the chromatogram. The intensity of the LipidII bands relative to the control is shown. **f** 2 nmol of mDAP containing LipidII was incubated with increasing mole:mole ratio of His-PknB-Ec$_{wt/mutant}$. The samples were extracted with BuOH/PyrAc. Addition of Trypsin to degrade His-PknB-Ec$_{wt}$ in the reaction mixture is indicated. The samples were extracted with organic solvent and resolved on TLC to analyze the presence of extractable LipidII. Source data are provided as a Source Data file

targets of PknB; and cluster 3 represents the proteins that are not the direct targets of PknB, wherein phosphorylation does not alter significantly upon depletion or complementation with the wild-type PknB. (Fig. 7e; Supplementary Table 4c). Interestingly, we observed that both cluster 2 and cluster 3 proteins were hyperphosphorylated by the mutant PknB-GM, exemplifying both specific as well as promiscuous hyperphosphorylation (Fig. 7e; Supplementary Table 4c).

**Hyperphosphorylation is a result of increased activity.** To confirm that mutant is indeed hyper-phosphorylated in the activation loop (Fig. 6), we raised phosphospecific antibodies that are capable of recognizing phosphorylated T171 and T173 residues (Fig. 8a). The specificity and sensitivity of the antibodies were characterized (Supplementary Fig. 5). Consistent with mass spectrometry data, we observed ~1.5-fold increase in the normalized activation loop phosphorylation in the

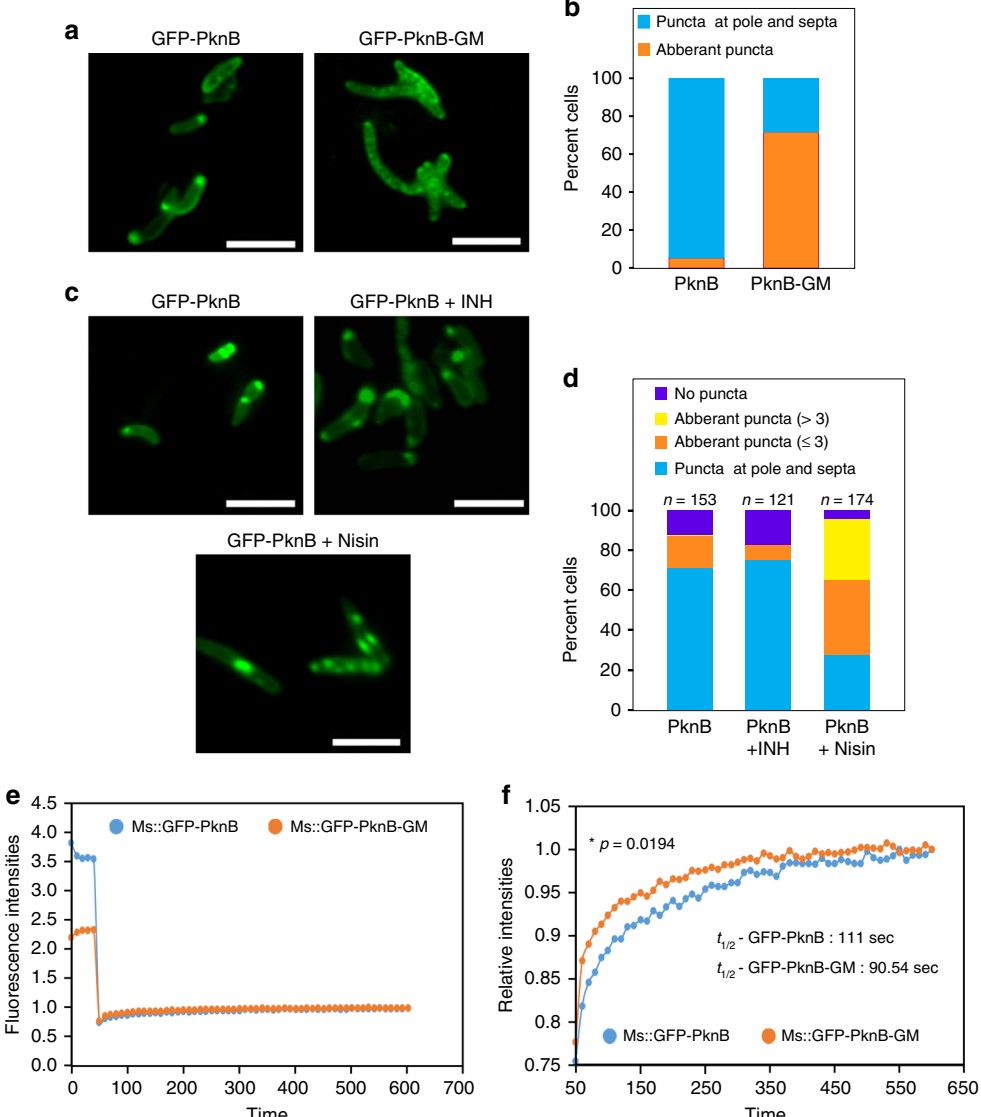

**Fig. 5** Abrogation of ligand binding perturbs localization of PknB. **a** *M. smegmatis pknB* conditional mutant (*mc²ΔB*) was electroporated with pNiT-GFP-PknB or pNit-GFP-PknB-GM constructs to generate *mc²ΔB::GFP-B* or *mc²ΔB::GFP-B-GM* strains. The strains were cultured in the presence of 50 ng per ml ATc and 0.2 µM IVN till $A_{600}$ of ~0.8. The cultures were washed thrice with $PBST_{80}$ to remove ATc and the cultures were grown for 6 hours in 7H9 media containing 1 µM IVN. Florescence images were captured using ×100 oil-immersion Zeiss Imager. M1 microscope. Scale bar- 5 µm. **b** 200 *mc²ΔB::gB* or *mc²ΔB::gB-GM* cell from **a** were analyzed for the localization of GFP-PknB$_{wt/mutant}$. The aberrant puncta collectively represents cells without any distinct localization or showing puncta at regions other than poles and septa. **c** *mc²ΔB::gB* strain cultured till $A_{600}$ of ~0.8 were washed and grown for 3 hours in 7H9 media containing 1 µM IVN in the presence or absence of 25 µg per ml nisin or 250 ng per ml INH. Scale bar- 5 µm. **d** Between 120 and 174 cells (as indicated) of *mc²ΔB::gB* cell from **c** were analyzed for the localization of GFP-PknB$_{wt}$. The no puncta phenotype has been mentioned as a distinct feature from aberrant puncta in this case for better analysis of phenotype. Scale bar- 5 µm. **e, f** *M. smegmatis mc²155* strain was electroporated with GFP-PknB or GFP-PknB-GM to generate *mc²::gB* or *mc²::gB-GM*. Cultures of *mc²::gB* or *mc²::gB-GM* grown in the presence of 0.2 µM IVN were used for the FRAP analysis. **e** The graph represents the mean fluorescence intensity plotted as a function of time by normalizing the intensity at each time point ($I_t$) to the first time point ($I_0$, $t = 0$ s) i.e., $I_t/I_0$. The values obtained were subtracted from the intensity at the time point of bleaching ($I_b$). **f** The time frames from 50 to 600 s are depicted, which highlight the difference in recovery times of GFP-PknB and GFP-PknB-GM. The $t_{1/2}$ values of recovery obtained are an indication of the time taken for half the maximal recovery after bleaching in three biologically different experiments. Statistical analysis was performed with the unpaired *t*-test using Graphpad software. Source data are provided as a Source Data file

ligand binding mutant PknB-GM (Fig. 8b). GarA has previously been demonstrated to be a robust in vitro substrate for PknA, PknB, and PknG[36], hence we performed *in vitro* kinase assays with immunoprecipitated 3F-PknB and 3F-PknB-GM using GarA as the substrate. It is apparent from the data that PknB-GM showed higher activity compared with the PknB (Fig. 8c, d), which could be a combinatorial effect of higher phosphorylation of loop as well as the juxtamembrane

residues. Even though GarA is a robust in vitro substrate for PknB, in in vivo it is majorly phosphorylated by PknG on T21 residue[36,37]. In agreement with this we observed that phosphorylation of GarA on T21 is unperturbed by depletion of PknB as well as complementation (Fig. 7e). On the other hand, phosphorylation of TatA on T60 was found to be PknB dependent, which showed hyperphosphorylation upon complementation with PknB-GM (Fig. 7e). In an independent

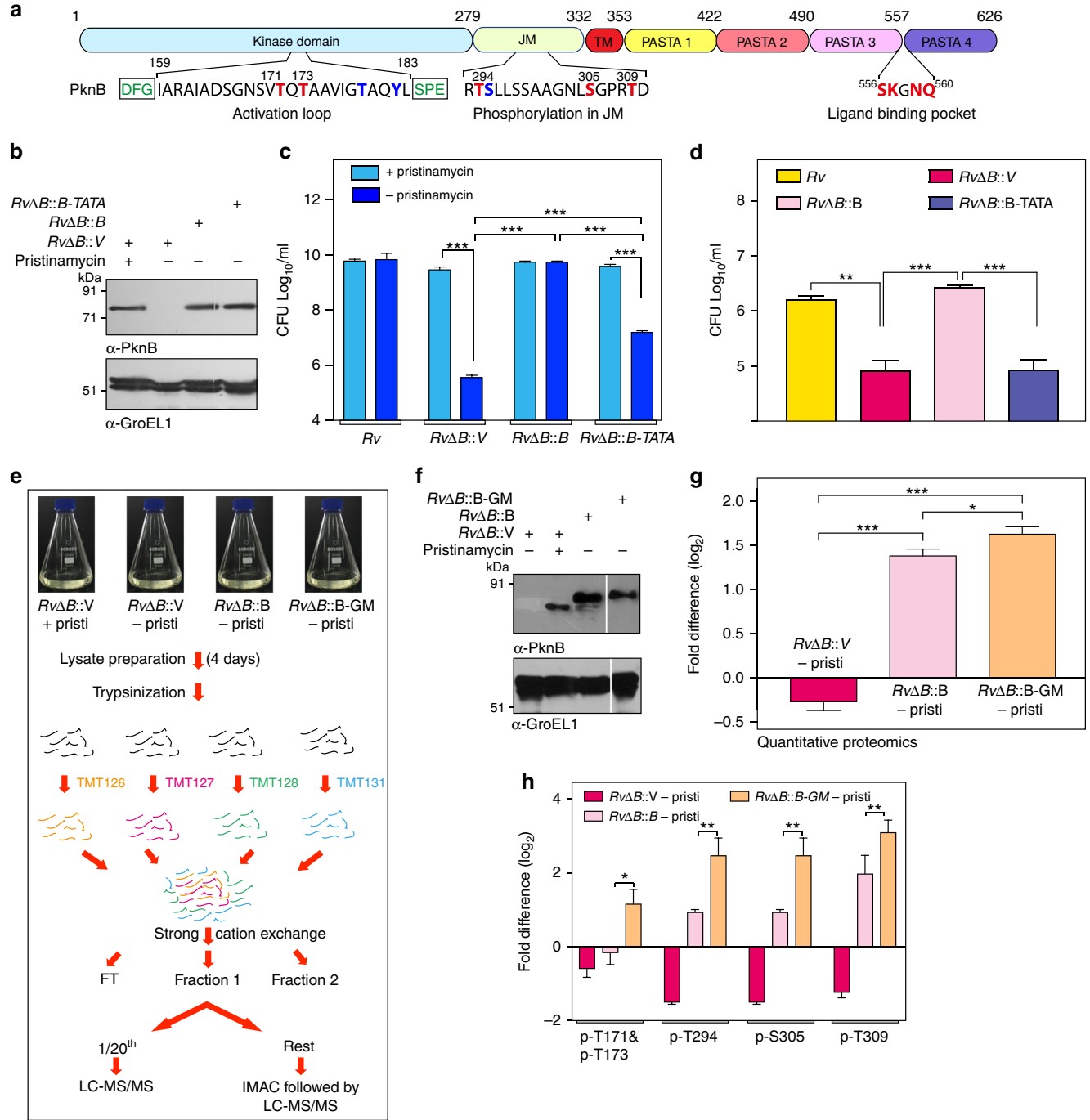

study, phosphorylation of TatA on T60 was shown to be dependent on PknA and PknB[34].

We sought to validate the data by quantitating the peak area in an independent mass spec experiment for phosphopeptides corresponding to GarA(T21) and TatA(T60). Depletion of PknB in the absence of inducer and expression of PknB and PknB-GM was confirmed by western blots (Fig. 8e). In concurrence with the TMT data (Fig. 7), phosphopeptide corresponding to GarA-T21 showed similar peak area in $Rv\Delta B$ & $Rv\Delta B$::B-GM samples, with slight decrease in $Rv\Delta B$::B sample (Fig. 8f). On the other hand, phosphopeptide corresponding to TatA showed distinct hyper-phosphorylation in $Rv\Delta B$::B-GM compared with $Rv\Delta B$ & $Rv\Delta B$::B (Fig. 8f). To further substantiate the data, we performed parallel reaction monitoring (PRM) to quantitate the TatA-T60

phosphopeptide (Supplementary Figs. 7 and 8g). Quantitation of TatA-T60 phosphopeptide with respect to the corresponding heavy peptide using PRM evidently demonstrated ~2-fold (31.2 fmoles) increase in its levels in $Rv\Delta B$::B-GM sample compared with $Rv\Delta B$ & $Rv\Delta B$::B (18.6 and 16.4 fmoles) samples. Collectively, these data demonstrate that the abrogation of ligand binding perturbs the normal regulatory circuits of PknB, resulting in aberrant localization, hyperactivation of the kinase, and indiscriminate target-specific and promiscuous phosphorylation events, leading to eventual cell death (Fig. 8h)

## Discussion

Since the domain structure of bacterial STPKs are similar to their eukaryotic counterparts[28,38,39], the hypotheses with respect to

**Fig. 6** PknB-GM is hyper-phosphorylated in the activation loop. **a** Threonine and tyrosine residues that are be known to be phosphorylated are highlighted by red or blue color. The phosphorylations that are identified in the current study are shown in bold red. The ligand binding residues are also indicated. **b** $Rv\Delta B$ strain was electroporated with integrating pST-CiT, pST-CiT-PknB, and pST-CiT-PknB-TATA to generate $Rv\Delta B::V$, $Rv\Delta B::B$, and $Rv\Delta B::B$-TATA, respectively. The transformanats grown in the presence of pristinamycin were seeded at $A_{600}$ of ~0.05 and grown for 5 days in the presence or absence of pristinamycin. WCLs were resolved on SDS-PAGE and probed with α-PknB and α-GroEL1 antibodies. **c** Cultures of $Rv$ and $Rv\Delta B$ transformants were inoculated at $A_{600}$ of ~0.05 and grown in the presence or absence of pristinamycin for 6 days and CFUs were enumerated. Data are representative of one of the three biological experiments and each experiment was performed in triplicates. **d** Cultures of $Rv$ and $Rv\Delta B$ transformants grown till $A_{600}$ of ~0.6–1.0 were washed thrice and used for infecting differentiated THP-1 cells (1:10 M.O.I). CFUs were enumerated at 72 h p.i. on plates containing pristinamycin. **e** Outline of the protocol used for TMT-based quantitative phosphoproteomics analysis. **f** $Rv\Delta B$ transformed with pNit-3F or pNit-3F-PknB or pNit-3F-PknB-GM were grown in presence of pristinamycin or IVN for four days and WCLs prepared were probed with α-PknB and α-GroEL1 antibodies. **g** The absolute PknB levels in depleted and complemented strains were assessed by quantitative TMT-based proteomics approach. Each data point is represented by TMT intensities obtained from three technical replicates. Data represents mean + SD. *$p < 0.05$, ***$p < 0.0005$. **h** The levels of phosphorylation of various residues in PknB activation loop and juxtamembrane regions were estimated from the TMT experiment. Phosphorylated peptides from three replicates normalized to their corresponding PknB protein intensities were considered for this analysis. The data is represented as log2 values and the raw data of the experiment is presented in Supplementary Table 1. Data represents mean + SD. Statistical analysis was performed with the unpaired $t$-test using Graphpad software. *$p < 0.05$, **$p < 0.005$, ***$p < 0.0005$. Source data are provided as a Source Data file

their activation and regulation are influenced by the findings in eukaryotic kinases. There are two major mechanisms by which the activity of a protein kinase is regulated: (a) by modulating protein expression levels and (b) by limiting the levels of activity through phosphorylation and dephosphorylation of the activation loop residues. In case of PknB, the following findings have cumulatively led to the formulation of the activation mechanism hypothesis presented in Fig. 1b: (i) $PknB_{Mtb}$ is autophosphorylated in the activation loop and this phosphorylation is necessary and sufficient for its activity in vitro[27], (Supplementary Fig. 4). (ii) $PknB_{Mtb}$ forms both back-to-back[15] and front-to-front PknB[14] dimers, and dimerization is a pre-requisite for activation loop phosphorylations[13] (iii). The PASTA domain interacts with mDAP-containing muropeptides, and this domain is adequate for appropriate localization of the protein[11]. Interestingly, PknB protein expression levels are downregulated during dormancy[9] and nutrient starvation[10] and are upregulated during exponential growth[8] and resuscitation[9], suggesting that PknB activity may also be regulated through the modulation of its expression pattern.

The PASTA domains across the bacterial kingdom share a highly conserved globular structure, although their sequences are diverse[40]. Various PASTA kinases have been demonstrated to harbor specific unique features. For example, a conserved arginine in PASTA3 has been shown to be a determining factor for ligand-binding in PrkC, the *Bacillus subtilis* PknB ortholog[41]. Similarly, three putative muropeptide binding sites in the hinge regions have been suggested to be the ligand binding pockets in StkP[42]. Recently, a unique citrate-binding site[19] and a hydrolase (LytB)-binding region[16] have been defined in the terminal PASTA domains of PknB and StkP, respectively, thus implicating other roles for the domain in addition to muropeptide binding. In *S. pneumoniae*, the terminal PASTA domain of StkP is both necessary and sufficient for its function[16]. In *Mtb*, we have previously reported that the terminal PASTA4 is absolutely essential for the function of PknB[5]. However, unlike in *S. pneumoniae*, we observe that the terminal PASTA4 is not sufficient; rather, appropriate length of the total PASTA domain is also vital (Fig. 1). The fact that PASTA4 plays an indispensible role is reinforced by the finding that SK and NQ residues in the PASTA3-4 linker region, serve as the putative ligand interacting residues (Fig. 2). In contrast to the tetra mutant that shows a drastic phenotype (Fig. 2), we observed varying levels of compromise in case of double mutants, with the data suggesting a greater role for mDAP-interacting residues compared with iGln-interacting residues (Fig. 3). Interestingly, both the double mutants as well the tetra mutant are similarly compromised in

their ability to functionally complement PknB depletion ex-vivo, suggesting that PknB plays a very stringent role at the time of infection and under these circumstances even minor perturbations are not tolerated (Figs. 2 and 3).

Muropeptides are widely acknowledged as the ligands for the PASTA domains of $PknB_{Mtb}$[11,12]. However, some experimental observations suggest that they may not be the primary ligand: for example, in vitro binding affinity of muropeptides to purified PknB-Ec is relatively weak, with micromolar concentrations of muropeptide being required for the binding[2,11,12], and the growth inhibition due to overexpression of PknB-Ec could not be ameliorated upon the addition of muropeptides mixtures[43]. Optimal binding occurs only when the MurNAc sugar and the stempentapeptide (muramyl pentapeptide) are present in the ligand. The other possible source of PknB ligand is peptidoglycan precursors such as LipidII, also present in the periplasmic space. LipidII possesses all the signatures of the muropeptide ligand and is also spatio-temporally localized to the same niche as PknB. In line with this, $PknB_{sa}$ was recently shown to bind very efficiently with LipidII molecules[22]. Our data also demonstrates efficient binding of PknB-Ec with mDAP-LipidII (Fig. 4). The weaker binding affinity compared to $PknB_{sa}$ could be either due to the fact that we tested only the PknB-Ec, or could be due to the lack of additional modifications (such as amidation), known to be present in mDAP as well as iGlu residues in *Mtb*[44]. Importantly, all three PknB-Ec mutants failed to interact with the mDAP-LipidII, clearly showing abrogation of ligand binding (Fig. 4).

The extracytoplasmic PASTA domain by itself has previously been shown to be sufficient for appropriate localization of PknB, which suggested that the interaction of this domain with the ligand may dictate the localization[11]. The data presented in Figs. 4 and 5 of this study suggest that LipidII may be the primary intracellular ligand, as incubating the cells with sublethal doses of the drug "nisin" results in an aberrant distribution pattern of PknB in the cell. Previous studies have shown that the addition of purified intracellular PASTA domain to the culture also inhibits the growth of *Mtb*[43]. Considering this fact with the data in Fig. 5, it is evident that incubating the cells with purified extracytoplasmic PASTA domain or nisin results in sequestration of available LipidII, thus compromising the functionality of PknB. The NMR structure of PASTA domain suggests that it is a linear domain. It is not clear as to how the muropeptide moiety in the LipidII interacts with the linear PASTA domain. It is possible that the interaction occur once the muropeptide is cleaved from the decaprenyl moiety. Alternatively, in the cellular context the PASTA domains may not be linear in structure. The regulation

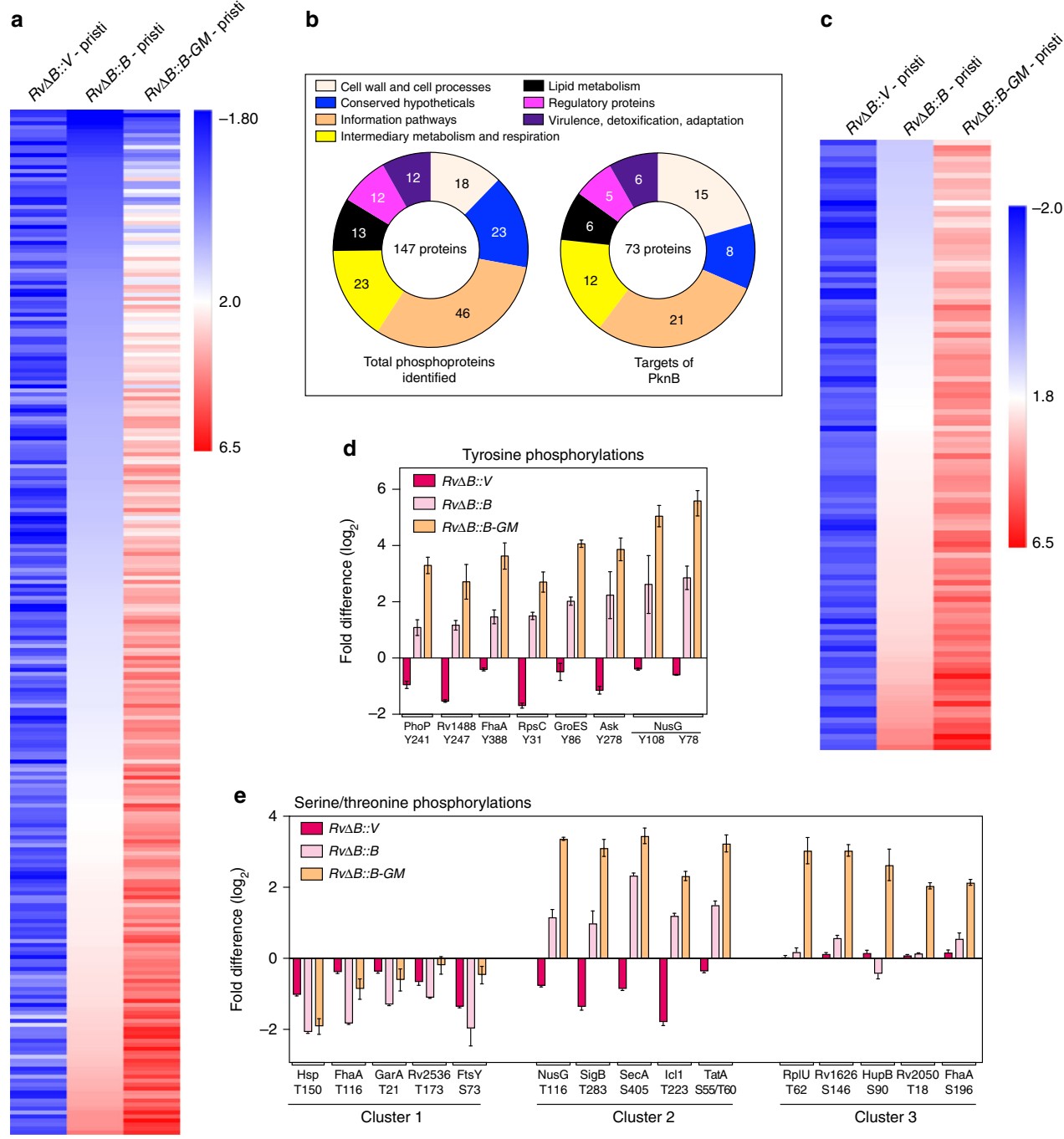

**Fig. 7** Ligand binding mutation causes global hyperphosphorylation of specific and non-specific targets. **a** TMT intensities of phosphopetides in $Rv\Delta B$ -pristinamycin (PknB depleted sample, labeled with TMT 127) or in $Rv\Delta B::B$ (complemented with 3F-PknB, labeled with TMT 128) or $Rv\Delta B::B\text{-}GM$ (complemented with 3F-PknB-GM, labeled with TMT131), with respect to $Rv\Delta B$ +pristinamycin as the reference comparator (control strain, labeled with TMT 126). The intensities of phosphopeptides in each case were normalized with respect to the corresponding absolute protein intensities and the values were converted to log2 values. Data were sorted with respect to $Rv\Delta B::B$ sample and heatmap of the data were generated using online tool Morpheus. **b** The 257 phosphopeptides detected in TMT experiment belonged to 147 unique proteins, which were classified according to their functional category with reference to mycobrowser database. The phosphopeptides were categorized as probable PknB substrates if the TMT $\log_2$ phosphointensity upon depletion was $<-0.32$ and upon complementation was $>1$. 111 phosphopeptides were classified as probable PknB substrates, which belonged to 73 unique proteins. The 73 PknB targets were also functionally characterized according to mycobrowser database. **c** Normalized TMT intensities of all 111 phosphopeptides which are probable PknB targets were converted to $\log_2$ values and data were sorted with respect to $Rv\Delta B::B$. Heatmap was generated using online tool Morpheus. **d** Normalized TMT intensities of 5 phosphopeptides each belonging to cluster 1, cluster 2, and cluster 3. **e** Normalized TMT intensities of PknB-dependent tyrosine phosphorylations are represented. Source data are provided as a Source Data file

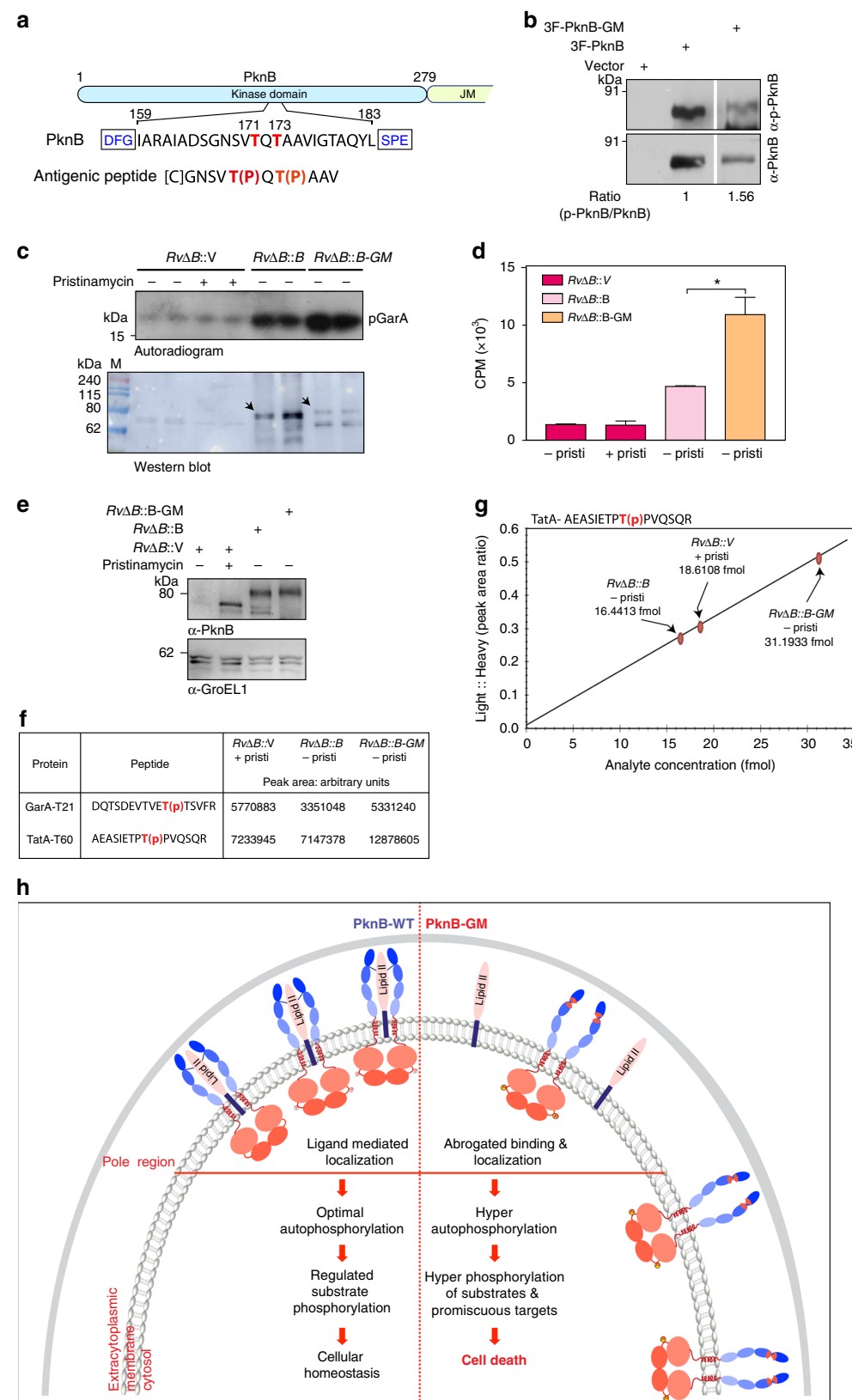

of the synthesis and continuous remodeling of PG is a fundamental process of the bacterial cell involving multiple proteins. PknB$_{Mtb}$ is a well-known modulator of multiple substrates involved in cell division and cell wall synthesis,

phosphorylating MviN and FhaA proteins among others, a molecular event that is linked to reduced PG biosynthesis[45,46]. Thus, a feedback loop regulation mechanism intertwining PknB activity and peptidoglycan synthesis may exist.

**Fig. 8** Hyperphosphorylation is a result of increased activity of the PknB-GM mutant. **a** Schematic outline showing the activation loop region of PknB and the peptide sequence used for generating phospho-specific antibodies. **b–d** WCL were prepared from $Rv\Delta B::V$ grown in the presence and absence of pristinamycin; $Rv\Delta B::B$ and $Rv\Delta B::B$-GM grown in the absence of pristinamycin and in the presence of IVN for 4 days. **b** 3F-PknB or 3F-PknB-GM proteins were immunoprecipitated from 1 mg WCLs using FLAG-M2 beads and 1/10th of IP was probed with α-PknB antibodies and 9/10th of IP was loaded for α-pPknB antibodies. The ratio of signal from total PknB and phospho-PknB was evaluated with the help of imageJ. **c** In vitro kinase assays samples were resolved on 15% SDS-PAGE, transferred to nitrocellulose membrane and autoradiographed. Upper part of the membrane corresponding to 40–240 kDa was probed with α-PknB antibody. **d** The bands corresponding to p-GarA (~17 kDa) were excised from the membrane described in **c** and counts per minute (CPM) were determined (right panel). **e** 5 mg WCL prepared from $Rv\Delta B::V$ +pristinamycin; $Rv\Delta B::B$ −pristinamycin+IVN and $Rv\Delta B::B$-GM -pristinamycin+IVN cultures grown for 4 days were digested with trypsin, and phosphoenriched. **f** We determined the peak area for GarA-T21 and TatA-T60 phosphopeptides using targeted proteomics. Table shows the total integrated area of GarA-T21 and TatA-T60 phosphopeptide in the samples. **g** Determination of absolute endogenous TatA-T60 phosphopeptide amounts in the samples by PRM. Graph showing absolute endogenous TatA-T60 light phosphopeptide concentration (fmoles) in phosphoenriched samples with respect to the standard curve generated by measuring peak area of different concentrations of heavy standard peptide (Supplementary Fig. 7). The horizontal axis represents Light: Heavy phosphopeptide integrated peak area ratios determined by skyline software. The vertical axis represents the absolute endogenous phosphopeptide TatA-T60 analyte concentration in fmoles. **h** Model depicting PknB regulation wherein LipidII interacts with specific region of PASTA 3-4 linker region of PknB and defines its localization to polar/septal niches and regulates the activity to optimal levels and hence maintains cellular homeostasis and cell survival. Source data are provided as a Source Data file

If the ligand binding is required for the activation of PknB, abrogation of binding should result in compromised loop phosphorylation. Contrary to this supposition, we observed significant hyperphosphorylation of activation loop and juxtamembrane regions in the PknB-GM mutant (Fig. 6). $PrkC_{bs}$ localizes to the division site and interacts and phosphorylates GpsB, which in turn regulates its activity by inhibiting its auto/transphosphorylations[47]. We hypothesize that upon binding of ligand, the kinase would be localized to the appropriate niche, whereupon a combination of other regulatory proteins/partners (including PstP) would ensure tight regulation of autophosphorylation levels, and by extension, kinase activity. In consonance, results (Fig. 8) showed that 3F-PknB-GM to be a more active kinase compared with 3F-PknB. The mis-regulated kinase results in hyperphosphorylation events targeting proteins, which are both canonical and non-canonical substrates that may be leading to aberrant functionality, leading to eventual death (Fig. 7). PRM analysis showed ~2-fold increase in the phosphorylation of the substrate of TatA on T60 residue upon complementation with PknB-GM, thus validating the data (Fig. 8). There are few examples in literature wherein phosphomimetic mutant of PknB substrates such as InhA, KasB, PcaA, and CwlM have been shown to have significant impact on the catalytic function and/or survival defects[46,48–51]. Complementation with phosphomimetic mutant of KasB (T334D/T336D) results in loss of acid fastness character and also causes loss of virulence[50]. Phosphomimetic mutant of InhA (T266E) fails to rescue $Msmeg$/$Mtb$ $inhA$ conditional mutant upon depletion[48,49]. PcaA-T168D/T183D phosphomimetic mutant shows reduced bacterial survival and defective mycolic acid profile[51]. Recently, a double phosphomimetic mutant of a major substrate of PknB, CwlM (T382D + T386D) was shown to be defective in complementing the mutant strain[46].

The hyperphosphorylation of the juxtamembrane domain could be linked to the hyperphosphorylation of the activation loop. Although the specific role of PknB juxtamembrane phosphorylation is unknown, it might be critical for transducing the ligand-mediated signal to the intracellular kinase domain, or for recruitment of regulatory interacting partners. Based on in vitro phosphorylation assays, Sassetti's group suggested that PknB and PknH are master regulators that are capable of phosphorylating multiple other kinases[6]. It is possible that promiscuous hyperphosphorylation could be due to aberrant activation of other STPKs by mislocalized PknB. Even though, we did not find any other STPKs in our final phosphoproteome, we cannot negate this prospect. These aspects need further investigation. Future PknB interactome studies may provide possible insights into how PknB activation loop and juxtamembrane phosphorylations are regulated.

## Methods

**Generation of plasmid constructs.** $pknB$ was amplified from the genomic DNA of $Mtb$ and the amplicon was cloned into NdeI-HindIII sites of pNit-1[52] and pNit-3F vectors to generate pNit-B and pNit-3F-B, respectively. $PknB$ upto transmembrane region (NdeI-SapI) and $PASTA$-234 (SapI-HindIII) were PCR amplified and the digested amplicons were ligated with NdeI-HindIII digested pNit-3F to generate PknB-234. Similarly, $pknB$-12 and $PASTA$-12 were ligated with pNit-3F to generate PknB-1212. PknB activation loop mutants and PASTA point mutants were generated by overlapping PCR. The codon optimized nucleotide sequence of superfolder $gfp$ was commercially synthesized from Genscript. $pknB$ or $pknB$-GM were amplified using specific primers containing SapI and HindIII sites, and $sgfp$ was amplified from pUC57-sGFP using specific primers containing NdeI and SapI sites. The amplicons were digested and ligated with pNit-3F vector digested with NdeI-HindIII to generate pNit-GFP-PknB and pNit-GFP-PknB-GM. The extracytoplasmic domain of $pknB_{wt/mutant}$ was PCR amplified using pNit-3F-$PknB_{wt/mutant}$ as the template and the amplicons were digested with BamHI-HindIII and cloned into corresponding sites in pET28a vector to generate pET-$pknB$-Ec, pET-$pknB$-Ec-G, pET-$pknB$-Ec-M, and pET-$pknB$-Ec-GM. Oligonucleotides and strains used in the study are described in Supplementary Material.

**Analysis of growth, isolation of lysates, and western blot.** $pknB$ conditional mutants, $mc^2\Delta B$ and $Rv\Delta B$ (Rv-pptr-B)[5,17] were electroporated with pNit, pNit-3F or pST-CiT[53] derived constructs (Supplementary Material). Transformants were grown in 7H9 media containing ADC (10%) and pristinamycin 1A (1.5 µg per ml; Molcon Corp) till $A_{600}$ reached ~1.0. To determine the ability of mutants to rescue growth, the cultures were washed thrice with equal volumes of $PBST_{80}$ (1X PBS with 0.05% tween 80), diluted to $A_{600}$ of ~0.05, and grown for 6 days in the presence or absence of 1.5 µg per ml pristinamycin or 0.2 µM isovaleronitrile (IVN). CFUs were enumerated after 6 days of growth. To evaluate the expression of 3X-FLAG tagged wild type and mutant PknB proteins, the transformants were grown in the absence of pristinamycin and presence of IVN for 5 days and probed with anti-PknB (α-PknB), anti-phosphoPknB (α-pPknB), or anti-GroEL1 (α-GroEL1) antibodies. α-PknB and α-GroEL1 antibodies were raised in rabbits and were used at 1:10,000 dilutions. The rabbit polyclonal phospho-specific antibodies were custom generated by PhosphoSolutions (Aurora, CO) using the antigenic peptide "[C]GNSVT(P)QT(P)AAV", a sequence derived from the PknB activation loop. For the α-pPknB blot the membrane was blocked with 5% BSA followed by overnight incubation with α-pPknB (1:250 dilution in 5% BSA) at 4 °C. Monoclonal α-FLAG M2 (Sigma-F1804) antibody was used at 1:2500 dilution.

**THP-1 infections.** $Rv$ or $Rv\Delta B$ transformants grown up to $A_{600}$ of ~0.8 in the presence of 1.5 µg per ml pristinamycin were washed once with $PBST_{80}$ and twice with PBS to remove $Tween_{80}$ and pristinamycin and the cells were passed through 27 G needle syringe to make them into single-cell suspension. Human monocytic THP1 cells (ATCC-TIB-202) were cultured in RPMI (Hyclone) supplemented with 10% FBS (Invitrogen). THP1 infections were performed with $5 \times 10^5$ cells seeded in 24-well plates that were differentiated with 10 nM PMA for 24 h. Differentiated cells were allowed to recover for 12 h prior infection with Mtb at 1:10 or 1:4 MOI[54]. The extracellular bacteria were removed 4 h post infection by washing the cells thrice with sterile PBS, and this was considered as the zero time point. 0.2 µM IVN was added in the media wherever $Rv\Delta B::B_{wt/mutant}$ transformants were used for the infection. Cells were lysed in 1 ml of 0.1% triton-X100 at 0 and 72 h post infection and CFUs were enumerated on 7H11 plates containing 1.5 µg per ml pristinamycin.

**Identification of the muropeptide binding site**. The coordinates of the PASTA domains of mycobacterial PknB were obtained from the NMR structure (2KUI)[18]. The two monomeric PASTA3-4 domains were subjected to protein-protein docking using HADDOCK web server[55,56]. Surface exposed hydrophobic residues – Met586, Val593, Val604, and Ile619 – were treated as active residues. To investigate possible binding modes of muropeptide to the dimeric model of PASTA3-4 domains, muropeptide (N-acetylglucosaminyl-N-acetylmuramyl-ala-iso-Gln-meso-diaminopimelic acid-ala-ala; Molecular formula: $C_{40}H_{67}N_9O_{21}$) was docked using AutoDock4[57]. Coordinates of the muropeptide were generated using the Babel program of Open Babel package[58] and PyMOL software was used for visualization (The PyMOL Molecular Graphics System, Version 1.5.0.4 Schrö-dinger, LLC). The ligand was allowed maximum possible flexibility with 27 rotatable bonds and Gasteiger atomic charges assigned to it. Coordinates of dimeric model of PASTA3-4 domains were kept rigid, and the ligand was allowed to explore the entire surface through the construction of a grid box, within the docking grid of $60 \times 45 \times 29\,\text{Å}^3$ with a grid spacing of 0.475. The docking simulation involved 27,000 generations, and population size of 150 in each genetic algorithm (GA) run. Energy evaluations were carried out 2,500,000 times, and one best individual was chosen from each iteration of the 250 Lamarkian search GA runs. The rate of genetic mutations and cross-overs were set to 0.02 and 0.8, respectively. The most stable binding mode of the ligand had a free energy of binding of −4.29 kcal per mol. The ligand-receptor interface on the PASTA dimer was defined as residues from each of the PASTA3-4 monomers, which had at least one atom within 5 Å of any atom of the ligand.

**Purification and circular dichroism**. *E. coli* BL21(DE3) (Stratagene) cells transformed with the appropriate recombinant plasmid (pET28a-*pknB*-Ec, pET28a-*pknB*-Ec-G, pET28a-*pknB*-Ec-M, and pET28a-*pknB*-Ec-GM) were grown in 1 L LB-medium (50 µg per ml kanamycin) at 37 °C. At an $A_{600}$ of 0.6, IPTG was added at a final concentration of 1 mM to induce expression of the recombinant proteins at 30 °C. The purification of hexa-His tagged protein was performed as described earlier[48]. Jasco J-815 spectopolarimeter was used for analyzing the ellipticity changes for each protein in far-UV (195–250 nm) wavelength range at 20 °C. The ellipticity changes were converted into MRE (Molar residual ellipticity) values and plotted against wavelength using SigmaPlot version 10.0.

**In vitro LipidII interaction studies**. In vitro LipidII binding assay was performed using purified $His_6$-PknB-Ec$_{wt}$ or PknB-Ec$_{mutants}$. $His_6$-PknB-Ec$_{wt}$ or PknB-Ec$_{mutants}$ were incubated with 2 nmol LipidII at molar ratios ranging from 0.5 to 8:1 (PknB:LipidII) in 50 mM Tris/HCl pH 7.0, 5 mM $MgCl_2$, 60 min at 30 °C[22]. The reaction mixture was extracted with an equal volume of butanol/pyridine acetate (2:1; vol:vol; pH 4.2) and analyzed by thin layer chromatography (TLC)[22]. For tryptic digestions, 25 µg trypsin (Gibco) were added thereafter, and the mixture was incubated at 37 °C for 60 min.

**Fluorescence microscopy**. $mc^2\Delta B$ strain was electroporated with pNit-GFP-PknB or pNit-GFP-PknB-GM and the colonies exhibiting optimal GFP fluorescence were chosen. The cultures supplemented with 50 ng per ml ATc and 0.2 µM IVN (for inducing sGFP-PknB) were initiated at $A_{600}$ of ~0.025. After 12 h of growth cultures were washed twice with PBST$_{80}$ and supplemented with 7H9 containing 1 mM IVN. The cultures were withdrawn at 3 and 6 h post depletion of native *pknB* and fixed with 4% PFA for microscopic analysis. The fixed cultures (5 ml) were washed twice with 1X PBS and re-suspended in 100 µl PBS and 5 µl of fixed cultures were placed under a coverslip on a glass slide and observed in epi-fluorescence microscope at ×100 oil immersion using 488 nm excitation wavelength. For antibiotic treatment experiment, washed cultures supplemented with 7H9 media containing 1 mM IVN in the presence of 25 µg per ml nisin or 250 ng per ml INH were grown for 3 h prior to fixation and microscopy.

**FRAP measurements**. Cultures of $mc^2$-155 electroporated with pNit-GFP-PknB or pNit-GFP-PknB-GM ($mc^2$::GFP-PknB or $mc^2$::GFP-PknB-GM) were seeded at $A_{600}$ of ~0.05 and expression of GFP-PknB/PknB-GM was induced with 1 µM of IVN for 16 h at 30 °C. For imaging, 100 µl of culture spotted on a glass bottom dish and low-melting agarose (1% in 7H9) was layered and allowed to settle and dry before FRAP measurements. For FRAP experiments, the bacilli containing the GFP signal were focused and one end of each bacteria (covering 30–50% area) was photo bleached using a 488 nm laser at 100% output power using a 3i vector system (3i, USA) mounted on the Olympus IX83 inverted microscope, controlled using the Slidebook 6.0 software. The fluorescence recovery was measured by time-lapse imaging in the GFP channel (488 nm excitation through a SpectraX light engine and emission filter of 510/20 nm), using the ×60 1.35 NA plan-apochromat lens. The images were acquired using a Cascade II EM-CCD camera (Photometrics, USA) with EM gain for a total duration of 10 min with 10 s interval between two successive frames[59]. To plot the fluorescence recovery kinetics, the intensity normalized at each time point ($I_t$) to the first time point ($I_0$, $t = 0$ s). The values obtained were subtracted from the intensity at the time point of bleaching ($I_b$) [= $(I_t/I_0) - I_b$], allowing the recovery to be recorded with respect to $I_b$. The polar regions of ~120 cells for each sample were photo-bleached at the fifth frame (50 s) and the fluorescence recovery was monitored up to 600 s from at least three

biologically independent experiments. The mean relative intensity obtained was plotted as a function of time (in seconds) to calculate the rate of recovery ($t_{1/2}$, the time taken to attain half of the maximum intensity) in the bleached areas of each bacterium and plotted using the Non-linear fit equation in Prism 6.0.

**Immunoprecipitation and in vitro kinase assays**. $Rv\Delta B$ strains electroporated with pNit-3F-PknB$_{WT/GM}$ were inoculated at $A_{600}$ of ~0.05 and were grown in the absence of pristinamycin and presence of 0.2 µM IVN for 72 h. Cultures were centrifuged at 4000 rpm for 5 min and the pellets were resuspended in PBSG (PBS containing 5% glycerol) at 1:3 (pellet weight in g:PBSG volume in ml) ratio. Cells were lysed using 0.1 mm zirconium beads (Biospec) for 10 1 min cycles with 1 min intervals on ice using Biospec minibeadbeater. The lysates were clarified at 13,000 rpm at 4 °C and the concentration of supernatant whole-cell lysates (WCLs) were estimated. 1 mg of WCL was used for FLAG immunoprecipitation (IP) using FLAG-M2 beads and the 3F-PknB or 3F-PknB-GM were eluted with 0.1 M glycine (pH 2.2) and the eluate was neutralized by addition of 1/10th vol 1 M Tris-HCl (pH 8.0). The IPed proteins were resolved on SDS-PAGE and subjected to western blotting. The western blots were probed with 1:10,000 & 1:250 dilution of α-PknB and α-pPknB antibodies. The ratio of phospho-PknB band to that of PknB was evaluated using image J[60]. The in vitro kinase assays were performed in 30 µl reaction volume containing 25 mM HEPES-NaOH, pH 7.4, 20 mM magnesium acetate, 20 mM $MnCl_2$, 1 mM DTT, 100 µM ATP, 10 µCi of [γ$^{32}$P]ATP, and 2 µg of GarA and 10 µl eluate from IP for 30 min at 30 °C. The reactions were stopped by adding 15 µl of 6X-SDS sample buffer followed by heating at 95 °C for 5 min Reactions were resolved on 15% SDS-PAGE, transferred to nitrocellulose membrane and autoradiographed. The bands corresponding to pGarA were excised, incubated overnight in the scintillation cocktail (spectrochem) and counts per minute (CPM) were determined using Perkin Elmer microbeta TriLux 1450 LSC & Luminescence counter.

**TMT labeling and MS/MS analysis**. Cultures pellets were resuspended in SDS lysis buffer (2% SDS, 50 mM Triethylammonium bicarbonate buffer (TEAB; sigma), PhosSTOP tablets (Roche)in PBSG (PBS containing 5% glycerol) at 1:3 (pellet weight in g:buffer) ratio. Samples were heated at 95 °C for 20 min followed by 10 cycles of beadbeating. The lysates were clarified and the concentration of supernatant WCLs were estimated with the help of BCA reagent (Pierce). WCL (250 µg) from each samples shown in Fig. 6e were reduced using 10 mM tris (2-carboxyethyl) phosphine at 55 °C for 1 h and alkylated using 10 mM iodoacetamide for 30 min at 25 °C. Samples were acetone precipitated and the pellet was resuspended in 100 µl of 100 mM TEAB and digested with 6 µg Trypsin (Promega) for 16 h at 37 °C. TMT labeling (Thermo Fisher Scientific) was performed as per manufacturer's instructions. Peptides from $Rv\Delta B$::$V$ +pristinamycin; $Rv\Delta B$::$V$ −pristinamycin; $Rv\Delta B$::$B$ - pristinamycin + IVN, and $Rv\Delta B$::$B$-GM - pristinamycin + IVN were labeled with 126, 127, 128, and 131 reporter ions respectively. Labeled samples were pooled, dried, and Strong Cation Exchange chromatography (SCX) was performed using two salt gradients, 100 and 350 mM KCl[61]. 1/20th of each fraction was secured for the total proteome analysis. Rest of the sample was enriched for the phosphopeptide using IMAC beads as described earlier[62].

The desalted samples for the total proteome analysis as well as the enriched peptide samples for phosphoproteomics analysis were reconstituted in Buffer A (95% Water, 5% Acetonitrile, 0.1% Formic acid). All experiments were performed using EASY-nLC system (Thermo Fisher Scientific) coupled to LTQ Orbitrap-Velos mass spectrometer (Thermo Fisher Scientific) equipped with nanoelectrospray ion source. A 10-cm PicoFrit Self-Pack microcapillary column (New Objective) was used to resolve the peptide mixture and the peptides were eluted at a flow rate of 300 nl per min for 120 min. The acetonitrile (containing 0.1% formic acid) gradient used for the run was 0–40% for 70 min, 40–80% for 10 min, 80% for 10 min, 80–0% for 5 min, and 0% for last 25 min LTQ orbitrap was used for the full MS scan. The peptides were dissociated with both HCD and CID for better MS/MS spectra. The collision energy induced dissociation of X ion precursors was performed at 35 for CID and 40 for HCD. Both MS and MS/MS data were acquired using scan range of 20–2000 M/Z ratios. The dynamic exclusion was set at 500 for both ion trap (CID) and FTMS (HCD) and the resolution was set at 7500. Spectra obtained were queried against *Mtb* H37Rv database (refseq database 85, release date 11 January 2018). Proteome Discoverer 1.3 was used as the search algorithm with oxidation of methionine and carbamidomethylation of cysteine as static modification. Phosphorylation of serine, threonine, and tyrosine was used as a dynamic modification. TMT 6 plex modification of peptide N-termini and lysine residues were set as the fixed modification. The TMT ratios were calculated with TMT126 ($Rv\Delta B$::$V$ +pristinamycin sample) as the reference comparator. All PSMs were identified at 1% false discovery rate (FDR). Mass tolerance for precursor ions and fragment ions were set at 10 ppm and 0.1 Da respectively. Mass-spectrometry analysis was performed to obtain a list of intensities of various proteins in proteome and phospho-enriched proteome. The phosphorylation sites in the protein for the phosphopeptides were identified with the help of in-house scripts tool for the proteomic analysis (http://protocols. ibioinformatics.org/InHouse/). A localization probability cutoff of 75% was defined in the in-house scripts proteomic analysis tool prior to p-site analysis. The p-sites hence identified were further analyzed for pRS score and PEP values. The p-sites with pRS score >50 and PEP <0.05 were only considered for further analysis. The

common proteins represented in the total proteome and phosphoproteome were analyzed using venn diagram generating tool venny[63]. The phosphointensity ratio of each phosphopeptide was normalized against the whole-protein intensity ratio of the corresponding protein. The normalized phosphointensity from three replicates was averaged and the average phosphointensity ratios were converted into $\log_2$ values. Heatmap was generated using online tool Morpheus (https://software.broadinstitute.org/morpheus/). The functional characterization of individual phosphoproteins identified in the study was done using mycobrowser database. The TMT-phosphoproteomics data has been submitted to the ProteomeXchange Consortium (http://proteomecentral.proteomexchange.org)[64] and can be accessed using data set identifier PXD012180 via the PRIDE partner repository.

**Targeted proteomics.** Synthetic isotopically labeled (SIL) peptides (Maxi Spike-Tides QL_AAA-peptides) with C-terminus $^{15}N$ and $^{13}C$ -labeled arginine were purchased from JPT Peptide Technologies GmbH (Berlin, Germany). The synthetic peptides were resuspended in 0.1 % Formic acid at a final concentration of 1.7 µg per µl and working concentrations of 0.6, 6, 60, and 600 fmol per µl were prepared. All the samples were analyzed using EASY-nLC 1000 system (Thermo Fisher Scientific) coupled to Thermo Fisher-Orbitrap Q- Exactive mass spectrometer (Thermo Scientific, Germany) equipped with nanoelectrospray ion source. Samples were loaded with buffer A and eluted with a 0–40% split gradient of buffer B (95% acetonitrile, 0.1% formic acid) at a flow rate of 300 nl per min for 30 min using 25 cm PicoFrit column. The acquisition method had a non-scheduled parallel reaction monitoring (PRM) event targeting the doubly charged precursor ion of the SIL peptides. The PRM event was performed with an orbitrap resolution of 17500 (at $m/z$ 200), a target AGC value of 1e6, and maximum fill times of 100 ms. Fragmentation was acquired with a normalized collision energy of 27 eV and MS/MS scan range of $m/z$ 100–1500 was used for mass determination. PRM data analysis was performed using skyline-Daily software version 4.1.1.18179 (https://skyline.ms/project/home/begin.view?). For reproducibility and precision of the PRM method, calibration graph was generated in skyline against different SIL peptide concentrations and most confident and intense transitions of the SIL peptide were used for quantification purpose. A fixed amount of SIL peptide (60 fmol) was spiked in each phospho-enriched sample (2 µg) for absolute concentration determination of endogenous phosphopeptide. The skyline file for data analysis has been attached as Supplementary Data 4, 5 and 6.

## Data availability

The TMT-phosphoproteomics raw data associated with Figs. 6 and 7 has been submitted to the ProteomeXchange Consortium (http://proteomecentral.proteomexchange.org) and can be accessed using data set identifier PXD012180 via the PRIDE partner repository. The Skyline files for the PRM data analysis (Fig. 8) are submitted as Supplementary Data 4–6. Images of unprocessed western blots, coomassie stained gels, TLC images, and autoradiograms used in the study are provided in Supplementary Fig. 8. The source data underlying Figs. 1a, f, 2d–e, 3c–g, 4c, 5b–d, 6c, d, 7d, e, and 8d are provided as a Source Data file. The source data underlying Fig. 6 are provided in Supplementary Table 1.

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

## Acknowledgements

This work was supported by the funding provided by Department of Biotechnology, Government of India (DST/INT/TUNISIA/P-17/2017 & BT/PR13522/COE/34/27/2015) to V.K.N.; P.K. is a Senior Research Fellow. We thank Dr. Francesca Forti for kindly gifting *RvΔB* conditional mutant. We thank the Central Mass Spec facility of at NII and Mrs. Shanta Sen for her support in managing the facility. We thank Dr. Swati Saha for critical reading of the manuscript. We are grateful to Dr. Sudeepa Rajan for her help with CD experiments and Dr. Savita Lochab, Dr. Mansoor Hussain for their assistance in microscopy.

## Author contributions

P.K., M.R., B.M., U.W., N.P.D., Y.C., S.S., and K.S. were involved in execution of experiments, data acquisition, analysis, and providing raw data for figures. T.S., G.D.J., D.S., D.M., F.G., and V.K.N. provided the scientific overview. P.K. and V.K.N. were involved in overall experimental design, manuscript writing, and making figures. V.K.N. guided the study.

## Additional information

**Competing interests:** The authors declare no competing interests.

