## [Peer Review File · Nature Communications]

Reviewers' comments:

Reviewer #1 (Remarks to the Author):

Tuberculosis is a leading cause of death worldwide, with an estimated one third of the world's population infected. Currently available intervention methods remain insufficient to control the global TB epidemic. It is therefore urgent that we gain new insights into molecular mechanisms mediating *M. tuberculosis* environmental adaptation. Within this context, protein phosphorylation is central for bacterial adaptation and in *M. tuberculosis*, Ser/Thr protein kinase (STPK) PknB is essential and vital for growth in culture. Kaur et al investigates the mechanisms of regulation of *M. tuberculosis* PknB, specifically the role of the kinase extracellular PASTA domains in the normal function of this kinase. The authors set out to answer several pertinent questions regarding kinase-ligand interactions including the identification of the ligand binding residues on the extracellular PASTA domain of PknB. The authors then, through a combination of mutations in the PASTA domains, assign functions to a specific domains and/or ligand binding interactions. Based on those mutations the study provides an exhaustive catalogue of important observations and inferences of PknB activity and PASTA ligand-interacting residues. The authors then performed phosphoproteomic analyses to investigate the global effect of abrogation of ligand binding to PknB on mycobacterial cells. Although this study is based on well-designed experiments, as detailed below, this reviewer has major concerns with the important missing/omitted data and the details regarding the phosphoproteomic analysis.

Major Points

1. The authors performed SCX sample fractionation and this was followed by Mass spec analysis using a LTQ Orbitrap-Velos. However the protein IDs are far too low when compared to what has been reported in literature for *Mtb*. As pointed further below, the claims and conclusions made throughout the text, specifically those regarding PknB hyperphosphorylation, should be based on an in depth analysis of both the proteome and phosphoproteome of the studied strains/conditions. The authors should provide more detailed information (even if it as supplementary material) about the MS scan parameters utilised (e.g. scan range, collision energy induced dissociation of X ion precursors, resolution, dynamic exclusion)

2. Please include the following in the mass spectrometry search parameters (page 12)

-# Release version/date of sequenced database

-Mass tolerance for precursor ions

-Mass tolerance for fragment ions

-Estimation of false discovery rate and how calculated

3. Compliance with spectral inspection of the p-sites. The authors mentioned that additional information regarding the identified p-site was up-loaded on “in-house scripts for proteomic analysis”. Nonetheless, it is not clear for this reviewer whether the authors performed any inspection of the annotated spectra or if the spectra of the p-sites identified were subject to further quality control (e.g. localization probability, PEP/Q values). In fact, considering the limited number of identified p-sites I would strongly recommend a manual inspection of the fragmentation spectra for good b and y-ion series. Finally I recommended that for ALL phosphopeptides identified, or annotated, mass-labeled MS/MS spectra need to be provided, either as supplementary material (as PDF or image files) or deposited in a public accessible repository such as PRIDE)

4. As mentioned above the limited number of protein IDs suggests a sub-optimal mass spec analysis and I would therefore suggest a further validation of the apparent differentially regulated p-sites. It would be of interest to confirm clusters 1, 2 and 3 by a Mass spec targeted approach MRM or PRMs of the some dysregulated p-sites.

5. Discussion regarding PknB target-specific and promiscuous phosphorylation events. In the discussion it is mentioned “In case of Mtb PknB it is possible that in the absence of ligand(s) the kinase is hyperphosphorylated, leading to loss regulation of its activity and promiscuous phosphorylation events targeting proteins that are not usual substrates, ultimately resulting in loss of cellular homeostasis, and eventual cell death.” This kind of statement would definitely benefit from a more in depth analysis.

The authors assume that hyperphosphorylated PknB is responsible for the abnormal phosphorylation of all “not usual substrates”, but I wonder if the authors have considered the possibility that hyperphosphorylated PknB phosphorylates any of the other Mtb STPKs. It is clear that one cannot exclude this possibility by the fact that none of the STPKs are listed in the limited list of phosphoproteins identified in this study. Additionally, cross phosphorylation between kinases occur in vitro as demonstrated by Robert Husson and co-workers, in which they show evidence that wild-type PknB phosphorylated kinase inactivated PknA. So it could well be the case that other STPKs are implicated in the phosphorylation of “not usual substrates”. It would be therefore of interest to investigate further if the p-sites distributed across the clusters (1,2 and 3) share a (PknB) phosphorylation site motif (see Prisic et al 2010) and assess the likelihood of these being phosphorylated in vivo by PknB and/or any other STPKs.

6. It is not clear how the phosphorylation of “not usual substrates” leads to eventual cell death. I suggest that the authors discuss this further by providing specific examples from the identified phosphoproteins/phosphopeptides that would justify such claims.

Others specific points:

1. M&M Analysis of growth isolation of lysates and western blot, they do not mention where the antibodies they use come from or their specificity

2. In immunoprecipitation (IP) of PknB-authors mention the use of image J to compare the ratio of phosphor-PknB band to the PknB, why was image J only used for this western blot and not all?

3. In the results section (iGln or mDAP interacting residues are independently essential for optimal PknB function, they state that the expression levels are similar, did they use Image J to confirm this? Is not clear from the text.
4. THP1 infections, there is no mention of washing off the bacterial after a certain time point. The method is referenced, references another paper. This seems like a ploy to bolster citation. Instead the original paper should be cited: Puri et al 2013 PLoS One.
5. M&M “THP1 infections were performed at 1:10 MOI as described earlier.” I am under the impression that MOI used in this study is too high in comparison with other studies that use 1:4, is there any explanation for this.

Reviewer #2 (Remarks to the Author):

This is a well written manuscript incorporating a large amount of generally high-quality data utilizing a pknB conditional depletion strain, complementation with mutated alleles, as well as several biochemical approaches. They first demonstrate binding of Mtb Lipid II to the Mtb extracytoplasmic domain which contains 4 PASTA repeats. These studies also confirm previously reported specificity for m-DAP vs. Lys containing muropeptides for PknB PASTA binding and the requirement for all 4 PASTA repeats for Mtb viability. Starting from computational docking studies suggesting that the second and third residues of the stem peptide of Mtb Lipid II interact with specific sites in the PASTA repeats, the authors provide evidence for the importance of 4 residues spanning the C-terminus of PASTA-3 and the N-terminus of PASTA-4, both for Lipid II binding in vitro and for viability of PknB in vitro and in THP-1 macrophage infections. The authors further demonstrate that PknB fails to localize to the mid-cell and poles in the pknB depletion strain complemented with constructs in which these residues are substituted with Ala, suggesting the importance of Lipid II binding for proper PknB localization. Finally, the authors perform quantitative mass spectrometry and identify hyperphosphorylation of both the juxtamembrane linker and the activation loop, as well as increased overall protein phosphorylation in the depletion strain complemented with the Ala-substituted binding residues in PASTA 3-4. Unfortunately, the phosphoproteomic data appear to be mis-annotated, so it is difficult to assess conclusions based on the data available.

This is an important contribution that will add substantially to our understanding of this essential kinase. It is not completely novel in that Lipid II was recently shown to be bound by the PASTA domains of the Staphylococcal orthologue of PknB, referenced by the authors. The identification of stem peptide interaction sites in PASTA-3 and PASTA-4 essential for cell viability and the increased activity of PknB with substitutions in specific residues required for binding are new and important findings. Though more would need to be done to know whether Lipid II is the in vivo PASTA ligand

vs. a means to deliver muropeptides to PknB PASTAs (see comments below), experiments to distinguish these possibilities would be extensive and beyond what would be a reasonable expectation for this manuscript that already contains a large amount of data. There are a few issues that the authors should address, most importantly reviewing and correcting the phosphoproteomic data, which appear to be mis-annotated. It is not clear how this will affect the interpretation of these data. Other issues can be addressed primarily by changes in text/discussion (see Comments for authors).

Specific Comments

Results:

1) The pristinamycin induction in the depletion strain achieves PknB levels that appear comparable to slightly higher than the H37Rv wild type levels, but the complemented strains expressing pNIT-regulated pknB with native and Ala-substituted PASTAs (- pristinamycin, + IVN) show PknB expression that appears to be ~2-4 fold higher than wild type (Figure 1D and Figure 6F). The functional assays (CFU) show good complementation despite this overexpression, but this may affect interpretation of the levels of phosphorylation in the mass spectrometry data, where native PknB complementation results in increased phosphorylation of many proteins, including known PknB substrates, and PknB-GM complementation results in even greater phosphorylation. The authors should address this issue in the text.

2) In the phosphoproteomic data, the authors should be clear what was used as the comparator to obtain the abundance ratios shown. Based on the schematic in Figure 6e and the graphs it appears that the depletion strain induced with pristinamycin is the comparator for the other strains, though this isn't actually stated in the figure legend or the text. Figure 1 indicates similar PknB levels in the pristinamycin-induced strain compared to wild type, so this is a reasonable comparison. To make the claim that there is hyperphosphorylation in the complemented strains, however, wild type H37Rv might have been a better comparison. The main claim is that the PknB-GM expressing strain shows greater phosphorylation than either the pristinamycin-induced strain or the native PknB complemented strain, which is supported by the data, but the magnitude of increased phosphorylation compared to physiologic phosphorylation in wild type is not known.

3) It was noted that although some previous PknB substrates were identified in Table 6a, individual phosphopeptides/phosphorylation sites did not match previous results. Checking a couple of specific proteins it appears that the phosphopeptides are mis-annotated. None of the three Rv2536 and none of the three Rv3246c peptides match a sequence in these proteins, but rather map to different Mtb proteins. I did not look at others in this table or other tables of the data, but the authors should review all of the proteomic data presented and correct this problem, and adjust their conclusions as appropriate.

4) Assuming the data are correct, surprising result in the quantitative phosphoproteomics data is that the complemented depletion strain (over)-expressing PknB does not show increased phosphorylation of the activation loop but does show increased phosphorylation of the juxtamembrane sites, whereas the complemented strain expressing PknB GM, which does not bind lipid II shows increased phosphorylation of both activation loop and juxtamembrane sites. Do the authors have an idea of how this might fit into their model?

Discussion

The first paragraph of the discussion is confusing. The authors state there are two major mechanisms for regulating kinase activity: expression and activation loop phosphorylation. For PknB they state that activation loop phosphorylation is necessary and sufficient for activation, that dimerization is required for this phosphorylation and that extracytoplasmic domain interacts with muropeptides which is required for localization. They then state that what has not been considered is 1) that the PknB kinase domain alone is active so that the extracellular domain is not required for activation and 2) that kinase expression levels are regulated under different conditions. Since many papers use the kinase domain alone, including dimerization and substrate phosphorylation papers, and the differential expression has been shown in several papers, it's hard to see why one would say these have not been considered. In this context, Figure 1b, which suggests ligand binding transmits a signal that activates PknB seems misleading. With the first structure of PknB published 15 years ago, it was evident that the disordered juxtamembrane segment was not likely to transmit a signal from extracytoplasmic ligand binding to the intracellular kinase domain, but rather that ligand binding was likely to be important for localization and/or dimerization (see Ref 37, Figure 3 and discussion).

In paragraph 2 the authors state that free muropeptides would not be present at sufficient concentration to function as in vivo EC-PknB ligands, yet their data indicate that it is stem peptide interactions of Lipid II, which presumably have similar binding affinities, that are required for lipid II binding. The local concentrations of lipid II or free muropeptides during growth is not known, but both may be high at sites of cell wall turnover. Though the data demonstrate clearly that PknB PASTAs can bind Lipid II it remains uncertain whether this is the primary in vivo ligand. Given the rigid linear structure of the EC domain, if the Lipid II bactoprenol is anchored in the cytoplasmic membrane could the stem peptide residues required for binding reach the PASTA 3-4 junction? If not do the authors imagine that the GlcNAc-MurNAc pentapeptide is released from bactoprenol moiety and that muropeptides are what is bound? Could Lipid II function to deliver sufficient concentrations of GlcNAc-MurNAc pentapeptide, rather than be the actual ligand? If Lipid II is indeed the ligand, could the bactoprenol moiety contribute to binding affinity?

While work to distinguish these possibilities is beyond the scope of this article will be needed to elucidate this, the authors could discuss these issues.

Minor comments

- a) The primary proteomic data should be deposited in an appropriate database, e.g. PRIDE
- b) Extracytoplasmic is preferable to extracellular. The PASTA domains of PknB are outside of the cytoplasmic membrane but internal to the “mycomembrane”.
- c) Page 2, paragraph 2, line 11. Incorrect reference (should be ref. 12)
- d) Page 4, end of first carryover paragraph. Add CD result here, since this is where the expression of the PknB GM protein is first described.
- e) Page 5. The FRAP experiments don't seem to add substantially to the results and could be deleted.
- f) Page 6, 2nd paragraph, line 2. PknB not PknG
- g) Page 11, 2nd paragraph. It would be appropriate to cite reference 23 which used this assay.
- h) Figure legends. State how P values were determined (only stated in Figure 1 legend)

Reviewer #1 (Remarks to the Author):

Tuberculosis is a leading cause of death worldwide, with an estimated one third of the world's population infected. Currently available intervention methods remain insufficient to control the global TB epidemic. It is therefore urgent that we gain new insights into molecular mechanisms mediating *M. tuberculosis* environmental adaptation. Within this context, protein phosphorylation is central for bacterial adaptation and in *M. tuberculosis*, Ser/Thr protein kinase (STPK) PknB is essential and vital for growth in culture. Kaur et al investigates the mechanisms of regulation of *M. tuberculosis* PknB, specifically the role of the kinase extracellular PASTA domains in the normal function of this kinase. The authors set out to answer several pertinent questions regarding kinase-ligand interactions including the identification of the ligand binding residues on the extracellular PASTA domain of PknB. The authors then, through a combination of mutations in the PASTA domains, assign functions to a specific domains and/or ligand binding interactions. Based on those mutations the study provides an exhaustive catalogue of important observations and inferences of PknB activity and PASTA ligand-interacting residues. The authors then performed phosphoproteomic analyses to investigate the global effect of abrogation of ligand binding to PknB on mycobacterial cells. Although this study is based on well-designed experiments, as detailed below, this reviewer has major concerns with the important missing/omitted data and the details regarding the phosphoproteomics analysis.

Major Points

1. The authors performed SCX sample fractionation and this was followed by Mass spec analysis using a LTQ Orbitrap-Velos. However the protein IDs are far too low when compared to what has been reported in literature for *Mtb*. As pointed further below, the claims and conclusions made throughout the text, specifically those regarding PknB hyper-phosphorylation, should be based on an in depth analysis of both the proteome and phosphoproteome of the studied strains/conditions. The authors should provide more detailed information (even if it as supplementary material) about the MS scan parameters utilized (e.g. scan range, collision energy induced dissociation of X ion precursors, resolution, dynamic exclusion)

We thank the reviewer for the in-depth review, which helped us in improving the manuscript. Till date there are four different phosphoproteomic studies in *Mtb* published by three groups. The first ever phosphoproteome study was published by Robert Husson's group in 2010 (Prisic et al. 2010). They reported identification of 500 phosphosites in 301 proteins, a sum total from six different experimental conditions. Warren's group in 2015 identified 214 proteins from 303 unique peptides from the exponential growth phase of a hypervirulent *Mtb* Beijing strain (Fortuin et al. 2015). Prasad's group reported identification of 512 phosphosites from 257 proteins, cumulative number of phosphosites obtained from exponential and stationary culture in *Mtb* H37Rv and H37Ra (Verma et al. 2017). In a recent study Robert Husson's group performed label free quantitative phosphoproteomics to identify direct targets of PknA and PknB. In this study they have identified 1241 phosphosites in 470 phosphoproteins (Carette et al. 2018).

In our study we performed TMT labeling to identify peptides that are differentially phosphorylated upon PknB depletion; complementation with wild type PknB and; complementation with PknB-GM. We identified a total of 242 phosphoproteins and 390 phosphopeptides common in three independent runs. We have in fact identified a cumulative total of 756 unique phosphopeptides three runs and 587 unique phosphopeptides in any two runs. The numbers are comparable to those obtained by Verma et al., wherein TMT labeling was performed.

However, we do acknowledge that the numbers were lower compared with those obtained by Dr. Husson's group in their recent study (Carette et al. 2018). We believe the following factors may have contributed to lower number of phosphopeptides and phosphoproteins identified in our study.

- a. TMT labeling may have decreased the coverage compared with the label free quantification.
- b. We have used Orbitrap Velos (older generation mass spec) with 10 cm column for LC compared with Orbitrap Q Exactive with 15 cm column for LC used by Husson's group.

As suggested by the reviewer, we have now provided extensive details of the analysis in the methods (Page 16, line 528-532).

2. Please include the following in the mass spectrometry search parameters (page 12)

- # Release version/date of sequenced database
- Mass tolerance for precursor ions
- Mass tolerance for fragment ions
- Estimation of false discovery rate and how calculated

As suggested we have provided the information in the Methods section (Page 16, line 533, line 537-539)

3. Compliance with spectral inspection of the p-sites. The authors mentioned that additional information regarding the identified p-site was up-loaded on "in-house scripts for proteomic analysis". Nonetheless, it is not clear for this reviewer whether the authors performed any inspection of the annotated spectra or if the spectra of the p-sites identified were subject to further quality control (e.g. localization probability, PEP/Q values). In fact, considering the limited number of identified p-sites I would strongly recommend a manual inspection of the fragmentation spectra for good b and y-ion series. Finally I recommended that for ALL phosphopeptides identified, or annotated, mass-labeled MS/MS spectra need to be provided, either as supplementary material (as PDF or image files) or deposited in a public accessible repository such as PRIDE)

Thanks for your comments. We have now provided all the details in the methods (page 16, line 543-546). Furthermore, we have manually analyzed the quality of spectra and found them to be good. We have also uploaded our data on PRIDE (PXD012180). Reviewers can access the data using the user ID: reviewer18517@ebi.ac.uk and password: iZEdZryY

During our initial p-site analysis using "in-house scripts for proteomic analysis", we had specified localization probability as >75% as a cut-off. However taking reviewer's comment seriously, we have further made our p-site analysis more stringent by only considering phosphosites with pRS score>50 and PEP<0.05. With these cut-offs the list of phosphopeptides for final consideration was reduced from 298 phosphosites to 257 phosphopeptides. We have modified the data represented in Fig 7 & supplementary mass spec files accordingly. We have also modified the numbers of phospho-sites and other details in results section.

4. As mentioned above the limited number of protein IDs suggests a sub-optimal mass spec analysis and I would therefore suggest a further validation of the apparent differentially regulated p-sites. It would be of interest to confirm clusters 1, 2 and 3 by a Mass spec targeted approach MRM or PRMs of the some dysregulated p-sites.

As suggested we have attempted targeted PRM approach to validate the data. Even though GarA is a robust *in vitro* substrate for PknB, *in vivo* it is majorly phosphorylated by PknG on T21 residue (O'Hare et al. 2008; Khan et al. 2017). In agreement with this data we observed that phosphorylation of GarA on T21 is unperturbed by depletion of PknB as well as complementation (Fig 7). On the other hand phosphorylation of TatA on T60 is PknB dependent, which shows hyperphosphorylation upon

complementation with PknB-GM (Fig 7). Two phosphopeptides corresponding to TatA were detected in our data. AEApSIETPpTPVQSQR showed phosphorylation at S55 and T60; and AEASIEETPpTPVQSQR showed phosphorylation only on T60 residue. Both the phosphosites were observed to be hyperphosphorylated by PknB-GM. However, we observed consistently higher intensity for the later peptide (T60) in our standardization experiments. Moreover phosphorylation on TatA-T60 was also reported in the previous studies (Prisic et al. 2010).

Thus based on the TMT data, previously published literature and standardization experiments, we selected two phosphopeptides: one corresponding to GarA-T21 whose phosphorylation does not get upregulated upon complementation with the PknB or PknB-GM (cluster 1) and the second corresponding to TatA-T60, which was hyperphosphorylated when complemented with PknB-GM. We did not select any peptides in cluster 3, as they were not consistently detected in the *RvAB* + pristinamycin samples during our standardization experiments.

Regrettably, upon receiving the peptide from JPT, we realized that by mistake they have synthesized a shorter heavy phosphopeptide, which was missing a Thr residue next to the phosphosite. Hence we decided to determine the peak area for the phosphopeptides corresponding GarA-T21 and TatA-T60. We started the experiment with four samples *RvAB* + pristinamycin, *RvAB* – pristinamycin (PknB depleted sample), *RvAB::B* – pristinamycin +0.2 uM IVN (complementation with the wild type) and *RvAB::B-GM* – pristinamycin +0.2 uM IVN (complementation with the ligand binding mutant). Unfortunately our sample preparation with *RvAB* – pristinamycin (PknB depleted sample) sample had technical issues and hence we were left with only three samples. Since the purpose of the experiment is to demonstrate no change or hyperphosphorylation of GarA-T21 or TatA-T60, respectively, upon complementation with PknB-GM compared with PknB or undepleted sample, we went on ahead to perform the experiment with these samples.

Western blot analysis confirmed depletion PknB in the absence of inducer and expression of PknB and PknB-GM in the complementation strains (Fig 8). In concurrence with the TMT data (Fig 7), phosphopeptide corresponding to GarA-T21 showed similar peak area in *RvAB* & *RvAB::B-GM* samples, with slight decrease in *RvAB::B* sample (Fig 8). On the other hand phosphopeptide corresponding to TatA showed distinct hyperphosphorylation in *RvAB::B-GM* compared with *RvAB* & *RvAB::B*. To further substantiate the data, we performed parallel reaction monitoring (PRM) to quantitate the amount of phosphopeptide corresponding to TatA-T60 using a synthetic heavy phosphopeptide (Sup Fig 7 & Fig 8f). Quantitation of TatA-T60 phosphopeptide with respect to the corresponding heavy peptide using PRM evidently demonstrated ~2 fold (31.2 fmoles) increase in the phosphopeptide levels in *RvAB::B-GM* sample compared with *RvAB* & *RvAB::B* (18.6 and 16.4 fmoles) samples.

5. Discussion regarding PknB target-specific and promiscuous phosphorylation events. In the discussion it is mentioned, “In case of Mtb PknB it is possible that in the absence of ligand(s) the kinase is hyperphosphorylated, leading to loss regulation of its activity and promiscuous phosphorylation events targeting proteins that are not usual substrates, ultimately resulting in loss of cellular homeostasis, and eventual cell death.” This kind of statement would definitely benefit from a more in depth analysis. The authors assume that hyperphosphorylated PknB is responsible for the abnormal phosphorylation of all “not usual substrates”, but I wonder if the authors have considered the possibility that hyperphosphorylated PknB phosphorylates any of the other Mtb STPKs. It is clear that one cannot exclude this possibility by the fact that none of the STPKs are listed in the limited list of phosphoproteins identified in this study. Additionally, cross phosphorylation between kinases occur in vitro as

demonstrated by Robert Husson and co-workers, in which they show evidence that wild-type PknB phosphorylated kinase inactive PknA. So it could well be the case that other STPKs are implicated in the phosphorylation of “not usual substrates”. It would be therefore of interest to investigate further if the p-sites distributed across the clusters (1, 2 and 3) share a (PknB) phosphorylation site motif (see Prisic et al 2010) and assess the likelihood of these being phosphorylated *in vivo* by PknB and/or any other STPKs.

We thank the reviewer for insightful comments. As suggested we have performed the motif analysis to determine the phosphosite distribution across all the clusters. Unfortunately, motif analysis neither provided new insights nor any consensus.

After applying relevant cutoffs we do not have any other STPKs in our final list. Based on *in vitro* phosphorylation assays, Sassetti’s group suggested that PknB and PknH are master regulators that are capable of phosphorylating multiple other kinases (Baer et al. 2014). We have shown in our previous manuscript that phosphorylation of PknA in the activation loop is independent of PknB (Nagarajan et al. 2015). Preliminary analysis by western blotting suggested that phosphorylation of PknA in the activation loop is not affected by wild type or mutant PknB complementation (data not included). However, we cannot ignore the possibility that mislocalized PknB may phosphorylate other kinases. We have modified our discussion to reflect these possibilities. In future, we aim to analyze these aspects in depth. (Page 12, line 404-408)

6. It is not clear how the phosphorylation of “not usual substrates” leads to eventual cell death. I suggest that the authors discuss this further by providing specific examples from the identified phosphoproteins/phosphopeptides that would justify such claims.

We believe that phosphorylation of “not usual substrates” alone may not be the reason responsible for cell death. We hypothesize that hyper-phosphorylation of both canonical as well as non-canonical substrates may be leading to aberrant functionality of these proteins, eventually leading to cell death.

There are few examples in literature wherein phosphomimetic mutant of substrates such as InhA, KasB, PcaA, CwlM have been shown to have significant impact on the catalytic function and/or survival defects. Complementation with phosphomimetic mutant of KasB (T334D/T336D) results in loss of acid fastness character and also causes loss of virulence (Vilcheze et al. 2014). Phosphomimetic mutant of InhA (T266E) fails to rescue *Msmeg/Mtb inhA* conditional mutant upon depletion (Khan et al. 2010; Molle et al. 2010). PcaA-T168D/T183D phosphomimetic mutant shows reduced bacterial survival and defective mycolic acid profile (Corrales et al. 2012). Recently a double phosphomimetic mutant of CwlM (major substrate of PknB) was shown to be defective in complementing the mutant strain (Turapov et al. 2018). We have modified our discussion to include the above aspects. (Page 12, line 388-400)

Others specific points:

1. M&M Analysis of growth isolation of lysates and western blot, they do not mention where the antibodies they use come from or their specificity.

Thank you for pointing this out. PknB and GroEL1 antibodies were raised in the lab. Usages of these antibodies have been reported in our previous manuscripts. However the point is well taken, and we have now included the details including the dilution used for all the antibodies used in this study (page 13, line 438-443).

- In immunoprecipitation (IP) of PknB-authors mention the use of image J to compare the ratio of phosphor-PknB band to the PknB, why was image J only used for this western blot and not all?

Image J analysis was performed for p-PknB analysis because we wanted to quantitate the difference in the activation loop phosphorylation. In other blots we did not perform quantitative analysis because our aim was only to establish expression of 3X-FLAG tagged PknB and PknB-mutant proteins in *RvΔB* upon depletion of pristinamycin inducible endogenous PknB. Sentence has been modified accordingly.

- In the results section (iGln or mDAP interacting residues are independently essential for optimal PknB function, they state that the expression levels are similar, did they use Image J to confirm this? It is not clear from the text.

We have not performed Image J quantitation for this blot. The purpose of the western blots here was to establish expression of 3F-PknB and 3F-PknB_{mut} proteins. Complementation experiments with 3F-PknB and 3F-PknB_{mut} have been done multiple times. The expression levels change from experiment to experiment, however the final result remains the same. What we presented in the manuscript was the best quality data, in which we have also enumerated CFUs. We have modified the statement in the manuscript accordingly.

- THP1 infections, there is no mention of washing off the bacterial after a certain time point. The method is referenced, references another paper. This seems like a ploy to bolster citation. Instead the original paper should be cited: Puri et al 2013 PLoS One.

The point is well taken. We apologize for not citing the original paper. The original reference has now been cited. The bacteria were washed off 4 h post infection. We have elaborated methods section (Page 13, line 448-449).

- M&M “THP1 infections were performed at 1:10 MOI as described earlier.” I am under the impression that MOI used in this study is too high in comparison with other studies that use 1:4, is there any explanation for this.

Many groups that work with *Mtb* routinely use 1:10 MOI for THP1 and RAW infection experiments. However, we agree the question is pertinent and hence we repeated the experiment at 1:4 MOI and the data is presented in Fig 3g (new figure). It is apparent from the data that changing MOI did not alter the results (Page 5, Line 153-155)

Reviewer #2 (Remarks to the Author):

This is a well-written manuscript incorporating a large amount of generally high-quality data utilizing a *pknB* conditional depletion strain, complementation with mutated alleles, as well as several biochemical approaches. They first demonstrate binding of *Mtb* Lipid II to the *Mtb* extracytoplasmic domain which contains 4 PASTA repeats. These studies also confirm previously reported specificity for m-DAP vs. Lys containing muropeptides for PknB PASTA binding and the requirement for all 4 PASTA repeats for *Mtb* viability. Starting from computational docking studies suggesting that the second and third residues of the stem peptide of *Mtb* Lipid II interact with specific sites in the PASTA repeats, the authors provide evidence for the importance of 4 residues spanning the C-terminus of PASTA-3 and the N-terminus of PASTA-4, both for Lipid II binding *in vitro* and for viability of PknB *in vitro* and in THP-1 macrophage infections. The authors further demonstrate that PknB fails to localize to the mid-cell and poles in the *pknB* depletion strain complemented with constructs in which these residues are substituted with Ala, suggesting the importance of Lipid II binding for proper PknB localization. Finally, the authors perform quantitative mass spectrometry and identify hyperphosphorylation of both the juxtamembrane linker and the activation loop, as well as increased overall protein phosphorylation in the depletion strain complemented with the Ala-substituted binding residues in PASTA 3-4. Unfortunately, the phosphoproteomic data appear to be mis-annotated, so it is difficult to assess conclusions based on the data available.

This is an important contribution that will add substantially to our understanding of this essential kinase. It is not completely novel in that Lipid II was recently shown to be bound by the PASTA domains of the Staphylococcal orthologue of PknB, referenced by the authors. The identification of stem peptide interaction sites in PASTA-3 and PASTA-4 essential for cell viability and the increased activity of PknB with substitutions in specific residues required for binding are new and important findings. Though more would need to be done to know whether Lipid II is the *in vivo* PASTA ligand vs. a means to deliver muropeptides to PknB PASTAs (see comments below), experiments to distinguish these possibilities would be extensive and beyond what would be a reasonable expectation for this manuscript that already contains a large amount of data. There are a few issues that the authors should address, most importantly reviewing and correcting the phosphoproteomics data, which appear to be mis-annotated. It is not clear how this will affect the interpretation of these data. Other issues can be addressed primarily by changes in text/discussion (see Comments for authors).

Specific Comments

Results:

1) The pristinamycin induction in the depletion strain achieves PknB levels that appear comparable to slightly higher than the H37Rv wild type levels, but the complemented strains expressing pNIT-regulated *pknB* with native and Ala-substituted PASTAs (- pristinamycin, + IVN) show PknB expression that appears to be ~2-4 fold higher than wild type (Figure 1D and Figure 6F). The functional assays (CFU) show good complementation despite this overexpression, but this may affect interpretation of the levels of phosphorylation in the mass spectrometry data, where native PknB complementation results in increased phosphorylation of many proteins, including known PknB substrates, and PknB-GM complementation results in even greater phosphorylation. The authors should address this issue in the text.

We thank the reviewer for the encouraging comments, insights, and thoughtful analysis regarding the mechanism. The manuscript has been revised in line with the reviewer's comments and we believe it has improved the quality of our manuscript considerably.

We agree that the induction of pNit-regulated 3F-PknB with 0.2 μ M IVN led to 2-4 fold hyper-expression as compared with the pristinamycin inducible PknB. However, the expression of complementation copies of PknB-WT and PknB-GM were comparable (Fig 6f and 6g). Hence the hyper

phosphorylation of activation loop, juxtramembrane residues, canonical and non-canonical substrates by PknB-GM seems to be due to higher kinase activity of mislocalized mutant.

In order to address these concern, we performed *in vitro* kinase assays with immunoprecipitated 3F-PknB and 3F-PknB-GM using GarA as the substrate. Even

though cpm readings obtained in duplicates were not normalized with respect to the protein levels (which were lower in case of PknB-GM), we observed PknB-GM to be hyperactive in our kinase assays. This data is included in the revised manuscript as Fig 8c & d. We performed another experiment wherein we analyzed the growth pattern on 7H11 plates in the presence or absence of pristinamycin or IVN as indicated (see the figure below).

Western blot analysis showed that in the absence of 0.2 μ M IVN the expression of wild type and mutant were lower compared with the pristinamycin inducible copy (Lane 2 vs 3 & 4). Despite the lower expression levels, 3F-PknB could successfully complement the growth phenotype (-pristinamycin; -IVN plate). On the contrary 3F-PknB-GM failed to complement the growth in the absence as well as in the presence of IVN. This data is included in the manuscript as Sup Fig 2b and c and is also mentioned in the text (page 5, Line 135-137). We attempted TMT labeling experiment with the lysates prepared from the above experiment to investigate the substrate phosphorylation profile. Unfortunately, phosphoenrichment in this experiment failed due to technical issues.

While it would be possible to modulate the levels of 3F-PknB and 3F-PknB-GM such that they are equivalent to endogenous PknB levels. This would require us to perform completely new set of experiments with different complementation strategy. However, the data from the above experiment suggests that the results are likely to be the same.

2) In the phosphoproteomic data, the authors should be clear what was used as the comparator to obtain the abundance ratios shown. Based on the schematic in Figure 6e and the graphs it appears that the depletion strain induced with pristinamycin is the comparator for the other strains, though this isn't actually stated in the figure legend or the text. Figure 1 indicates similar PknB levels in the pristinamycin-induced strain compared to wild type, so this is a reasonable comparison. To make the claim that there is hyperphosphorylation in the complemented strains, however, wild type H37Rv might have been a better comparison. The main claim is that the PknB-GM expressing strain shows greater phosphorylation that either the pristinamycin-induced strain or the native PknB complemented strain, which is supported by the data, but the magnitude of increased phosphorylation compared to physiologic phosphorylation in wild type is not known.

We apologize for not stating things clearly. We used *RvΔB* + pristinamycin (conditional depletion strain) labeled with TMT126 as the reference comparator. All the other samples, namely *RvΔB* - pristinamycin; *RvΔB::B* - pristinamycin and *RvΔB::B-GM* - pristinamycin were compared with respect to

the reference comparator. We have modified the methods and legend to bring out this aspect clearly (page 16, line 536-539; page 21, line 726-727). The levels of PknB in *Rv* and *RvΔB* are comparable and hence we think that using *RvΔB* + pristinamycin as the reference comparator may not be major issue in this experiment. Unfortunately, we did not perform the TMT labelling experiment with *Rv* and hence we are not in a position to provide data with *Rv* as the comparator.

3) It was noted that although some previous PknB substrates were identified in Table 6a, individual phosphopeptides/phosphorylation sites did not match previous results. Checking a couple of specific proteins it appears that the phosphopeptides are mis-annotated. None of the three *Rv2536* and none of the three *Rv3246c* peptides match a sequence in these proteins, but rather map to different *Mtb* proteins. I did not look at others in this table or other tables of the data, but the authors should review all of the proteomic data presented and correct this problem, and adjust their conclusions as appropriate.

We thank the reviewer for pointing this out. We regrettably state that there was a mistake while copying the data from Excel sheet (Sup table 3) to the word file (Sup table 6). In light of the comments from both the reviewers, we have reanalyzed the TMT data by making our p-site analysis more stringent considering phosphosites with pRS score > 50 and PEP < 0.05. We have carefully annotated the results to make sure that there are no mistakes.

4) Assuming the data are correct, surprising result in the quantitative phosphoproteomics data is that the complemented depletion strain (over)-expressing PknB does not show increased phosphorylation of the activation loop but does show increased phosphorylation of the juxtamembrane sites, whereas the complemented strain expressing PknB GM, which does not bind lipid II shows increased phosphorylation of both activation loop and juxtamembrane sites. Do the authors have an idea of how this might fit into their model?

We agree with the reviewer that this observation was rather surprising. It is true that we have observed increased phosphorylation of both activation loop and juxtamembrane residues in case of mutant and only in the juxtamembrane residues in case of wild type. Interestingly, *in vitro* kinase assays with immunoprecipitated 3F-PknB and 3F-PknB-GM demonstrated (Fig 6j in the revised manuscript), higher kinase activity for 3F-PknB-GM compared with the 3F-PknB.

We do not have a clear explanation for these observations. We hypothesize that upon binding of ligand, the kinase would be localized to the appropriate niche of the cell whereupon a combination of other regulatory proteins/partners (which might include the sole phosphatase PstP) would ensure tight regulation of autophosphorylation levels, and by extension, kinase activity. The interacting proteins at the pole and septum may be involved in monitoring and modulating the extent of phosphorylation in the activation loop and juxtamembrane residues. In case of PknB-GM mutant, the localization is disrupted and hence the interactions with the regulatory complex at the poles and septum. This in turn results in uncontrolled phosphorylation in the loop, juxtamembrane and hence hyperactivation of the kinase. Consequence of hyperactivation and mislocalization of the ligand-binding mutant is hyperphosphorylation of both canonical and non-canonical substrates. In future we aim to determine the role of juxtamembrane phosphorylations on the functionality of PknB. We aim to complement *RvΔB* with PknB, PknB-juxtamembrane phosphoablative, PknB-juxtamembrane phosphomimetic mutants to be able to determine their role in activation and protein-protein interactions.

Discussion

The first paragraph of the discussion is confusing. The authors state there are two major mechanisms for regulating kinase activity: expression and activation loop phosphorylation. For PknB they state that activation loop phosphorylation is necessary and sufficient for activation, that dimerization is required for this phosphorylation and that extracytoplasmic domain interacts with muropeptides which is required for localization. They then state that what has not been considered is 1) that the PknB kinase domain alone is active so that the extracellular domain is not required for activation and 2) that kinase expression levels are regulated under different conditions. Since many papers use the kinase domain alone, including dimerization and substrate phosphorylation papers, and the differential expression has been shown in several papers, it's hard to see why one would say these have not been considered. In this context, Figure 1b, which suggests ligand binding transmits a signal that activates PknB seems misleading. With the first structure of PknB published 15 years ago, it was evident that the disordered juxtamembrane segment was not likely to transmit a signal from extracytoplasmic ligand binding to the intracellular kinase domain, but rather that ligand binding was likely to be important for localization and/or dimerization (see Ref 37, Figure 3 and discussion).

We have gone through this paragraph carefully, and we agree that the way we had put forward argument is rather confusing. We have modified the first paragraph by deleting few sentences.

In paragraph 2 the authors state that free muropeptides would not be present at sufficient concentration to function as *in vivo* EC-PknB ligands, yet their data indicate that it is stem peptide interactions of Lipid II, which presumably have similar binding affinities, that are required for lipid II binding. The local concentrations of lipid II or free muropeptides during growth is not known, but both may be high at sites of cell wall turnover. Though the data demonstrate clearly that PknB PASTAs can bind Lipid II it remains uncertain whether this is the primary *in vivo* ligand.

We agree with the argument given by the reviewer. We have deleted a sentence, which talk about concentrations, as the concentrations of ligands are not known. We also agree with the reviewer that at this point of time we do not know if Lipid II is the primary ligand. Our localization experiments in the presence of Nisin seem to suggest that Lipid II may be the intracellular ligand. However, we do understand that these results are not the definitive proof that Lipid II is indeed the only ligand for PknB.

Given the rigid linear structure of the EC domain, if the Lipid II bactoprenol is anchored in the cytoplasmic membrane could the stem peptide residues required for binding reach the PASTA 3-4 junction?

Valid question. We have no idea about the orientation of PASTA domains in the cell. Are they standing upright or if they are bent a little?

If not do the authors imagine that the GlcNAc-MurNAc pentapeptide is released from bactoprenol moiety and that muropeptides are what is bound? Could Lipid II function to deliver sufficient concentrations of GlcNAc-MurNAc pentapeptide, rather than be the actual ligand? If Lipid II is indeed the ligand, could the bactoprenol moiety contribute to binding affinity? While work to distinguish these possibilities is beyond the scope of this article will be needed to elucidate this, the authors could discuss these issues.

All the above questions are spot on. Unfortunately, we do not have answers for any of them. Some aspects can be tested in future by performing *in vitro* binding experiments with Lipid II and mucopeptides to determine relative affinities. Thank you for understanding that the work to distinguish these possibilities is beyond the scope of this article. As suggested, we would discuss some of these issues in the discussion. (Page 11, line 371-374)

Minor comments

a) The primary proteomic data should be deposited in an appropriate database, e.g. PRIDE

We have deposited the primary data in PRIDE database and the reference number is provided in the revised manuscript (PXD012180).

b) Extracytoplasmic is preferable to extracellular. The PASTA domains of PknB are outside of the cytoplasmic membrane but internal to the “mycomembrane”.

We agree with the reviewer. We have replaced “extracellular” with “extracytoplasmic”

c) Page 2, paragraph 2, line 11. Incorrect reference (should be ref. 12).

We apologize-it is indeed incorrect reference. It has been corrected.

d) Page 4, end of first carryover paragraph. Add CD result here, since this is where the expression of the PknB GM protein is first described.

We have moved the CD results as suggested.

e) Page 5. The FRAP experiments don't seem to add substantially to the results and could be deleted.

It is true that it does not add substantially add to the results, FRAP data demonstrates that PknB-GM mutant has much higher dynamicity, thus laying foundation for the data presented in Fig 6 and 7. We would appreciate if we can retain this data in the manuscript.

f) Page 6, 2nd paragraph, line 2. PknB not PknG.

Thank you for noticing this mistake. It has now been corrected.

g) Page 11, 2nd paragraph. It would be appropriate to cite reference 23 which used this assay.

We agree with reviewer. Reference has been added.

h) Figure legends. State how P values were determined (only stated in Figure 1 legend)

We have determined P values the same way using the same program (Graphpad Prism). We have added the P value determination to all the legends in the revised manuscript.

REFERENCES

- Baer CE, Iavarone AT, Alber T, Sassetti CM. 2014. Biochemical and Spatial Coincidence in the Provisional Ser/Thr Protein Kinase Interaction Network of Mycobacterium tuberculosis. *J Biol Chem*.
- Carette X, Platig J, Young DC, Helmel M, Young AT, Wang Z, Potluri LP, Moody CS, Zeng J, Priscic S et al. 2018. Multisystem Analysis of Mycobacterium tuberculosis Reveals Kinase-Dependent Remodeling of the Pathogen-Environment Interface. *MBio* **9**.
- Corrales RM, Molle V, Leiba J, Mourey L, de Chastellier C, Kremer L. 2012. Phosphorylation of mycobacterial PcaA inhibits mycolic acid cyclopropanation: consequences for intracellular survival and for phagosome maturation block. *J Biol Chem* **287**: 26187-26199.
- Fortuin S, Tomazella GG, Nagaraj N, Sampson SL, Gey van Pittius NC, Soares NC, Wiker HG, de Souza GA, Warren RM. 2015. Phosphoproteomics analysis of a clinical Mycobacterium tuberculosis Beijing isolate: expanding the mycobacterial phosphoproteome catalog. *Frontiers in microbiology* **6**: 6.
- Khan MZ, Bhaskar A, Upadhyay S, Kumari P, Rajmani RS, Jain P, Singh A, Kumar D, Bhavesh NS, Nandicoori VK. 2017. Protein kinase G confers survival advantage to Mycobacterium tuberculosis during latency-like conditions. *J Biol Chem* **292**: 16093-16108.
- Khan S, Nagarajan SN, Parikh A, Samantaray S, Singh A, Kumar D, Roy RP, Bhatt A, Nandicoori VK. 2010. Phosphorylation of enoyl-acyl carrier protein reductase InhA impacts mycobacterial growth and survival. *J Biol Chem* **285**: 37860-37871.
- Molle V, Gulten G, Vilcheze C, Veyron-Churlet R, Zanella-Cleon I, Sacchettini JC, Jacobs WR, Jr., Kremer L. 2010. Phosphorylation of InhA inhibits mycolic acid biosynthesis and growth of Mycobacterium tuberculosis. *Mol Microbiol* **78**: 1591-1605.
- Nagarajan SN, Upadhyay S, Chawla Y, Khan S, Naz S, Subramanian J, Gandotra S, Nandicoori VK. 2015. Protein kinase A (PknA) of Mycobacterium tuberculosis is independently activated and is critical for growth in vitro and survival of the pathogen in the host. *J Biol Chem* **290**: 9626-9645.
- O'Hare HM, Duran R, Cervenansky C, Bellinzoni M, Wehenkel AM, Pritsch O, Obal G, Baumgartner J, Vialaret J, Johnsson K et al. 2008. Regulation of glutamate metabolism by protein kinases in mycobacteria. *Mol Microbiol* **70**: 1408-1423.
- Priscic S, Dankwa S, Schwartz D, Chou MF, Locasale JW, Kang CM, Bemis G, Church GM, Steen H, Husson RN. 2010. Extensive phosphorylation with overlapping specificity by Mycobacterium tuberculosis serine/threonine protein kinases. *Proc Natl Acad Sci U S A* **107**: 7521-7526.
- Turapov O, Forti F, Kadhim B, Ghisotti D, Sassine J, Straatman-Iwanowska A, Bottrill AR, Moynihan PJ, Wallis R, Barthe P et al. 2018. Two Faces of CwlM, an Essential PknB Substrate, in Mycobacterium tuberculosis. *Cell Rep* **25**: 57-67 e55.
- Verma R, Pinto SM, Patil AH, Advani J, Subba P, Kumar M, Sharma J, Dey G, Ravikumar R, Buggi S et al. 2017. Quantitative proteomic and phosphoproteomic analysis of H37Ra and H37Rv strains of Mycobacterium tuberculosis. *J Proteome Res*.
- Vilcheze C, Molle V, Carrere-Kremer S, Leiba J, Mourey L, Shenai S, Baronian G, Tufariello J, Hartman T, Veyron-Churlet R et al. 2014. Phosphorylation of KasB regulates virulence and acid-fastness in Mycobacterium tuberculosis. *PLoS Pathog* **10**: e1004115.

Reviewer #1 (Remarks to the Author):

The authors have satisfactorily addressed all my points listed previously and made the necessary changes in the manuscript.

Dr. Nelson Alexandre da Cruz Soares

Nelson C Soares (Ph.D.)

Applied Proteomics & Chemical Biology

Department of Integrative Biomedical Sciences

Health Sciences Campus

University of Cape Town

South Africa

*** Reviewer #2 (Remarks to the Author):

The authors have made substantial improvements in the manuscript since the initial review, including performing additional experiments and data analyses and changing the text in several places to improve clarity and accuracy.

Responses to major and minor comments from initial review are appropriate as noted briefly below.

Initial comment 1 (concern regarding overexpression of strains from pNIT promoter): The additional data shown in the authors' response and in figures 6 and 8 in the revised text address this concern and support their contention that PknB-GM is hyperactive kinase, and that increased phosphorylation by this strain is not the result of overexpression of this allele.

Initial comment 2 (what is the comparator for phosphorylation in different strains?): The authors have clarified which strain is the reference to which the others are compared. Though I would have preferred to see wild type as the reference, the authors point out that the depletion strain induced with ptc has similar levels of PknB compared to wild type, so the use of this strain is acceptable.

Initial comment 3 (Incorrect labeling of data in table (now suppl table 3): The data in the tables now appear to be appropriately labeled. Additionally, although the very large ratios are striking and don't need statistical confirmation, for those that are not so striking, adjusted P values would be useful.

Initial comment 4 (difference in PknB phosphorylation by PknB vs. PknB GM): This was an interesting observation, not requiring new data or revision. The authors discuss possible explanations and future plans that may shed light on this finding.

Initial comments on discussion (First paragraph not clear and discussion does not place their data in context appropriately): The revised first paragraph is much clearer and appropriately incorporates existing data regarding PknB activation. The discussion of the specific issues raised is much improved.

Responses to minor comments are appropriate. it is reasonable to include the FRAP data.

REVIEWERS' COMMENTS:

Reviewer #1 (Remarks to the Author):

The authors have satisfactorily addressed to all my points listed previously and made the necessary changes in the manuscript

Dr. Nelson Alexandre da Cruz Soares
Nelson C Soares (Ph.D.)
Applied Proteomics & Chemical Biology
Department of Integrative Biomedical Sciences
Health Sciences Campus
University of Cape Town
South Africa

We thank Dr. Nelson Alexandre da Cruz Soares for critical and insightful comments that helped us in improving the quality and presentation of the manuscript.

Reviewer #2 (Remarks to the Author):

The authors have made substantial improvements in the manuscript since the initial review, including performing additional experiments and data analyses and changing the text in several places to improve clarity and accuracy.

Responses to major and minor comments from initial review are appropriate as noted briefly below. Initial comment 1 (concern regarding overexpression of strains from pNIT promoter): The additional data shown in the authors' response and in figures 6 and 8 in the revised text address this concern and support their contention that PknB-GM is hyperactive kinase, and that increased phosphorylation by this strain is not the result of overexpression of this allele.

Initial comment 2 (what is the comparator for phosphorylation in different strains?): The authors have clarified which strain is the reference to which the others are compared. Though I would have preferred to see wild type as the reference, the authors point out that the depletion strain induced with *ptc* has similar levels of PknB compared to wild type, so the use of this strain is acceptable.

Initial comment 3 (Incorrect labeling of data in table (now suppl table 3): The data in the tables now appear to be appropriately labeled. Additionally, although the very large ratios are striking and don't need statistical confirmation, for those that are not so striking, adjusted P values would be useful.

We performed unpaired t-test analysis for the values in Supplementary Table 4 between the wild type complemented and ligand binding mutant complemented strains. The p values are now provided in the modified Supplementary Table 4.

Initial comment 4 (difference in PknB phosphorylation by PknB vs. PknB GM): This was an

interesting observation, not requiring new data or revision. The authors discuss possible explanations and future plans that may shed light on this finding.

Initial comments on discussion (First paragraph not clear and discussion does not place their data in context appropriately):The revised first paragraph is much clearer and appropriately incorporates existing data regarding PknB activation. The discussion of the specific issues raised is much improved.

Responses to minor comments are appropriate. It is reasonable to include the FRAP data.

We thank Reviewer 2 for in depth review, which helped in reorganizing the text that improved the clarity of presentation. Thank you for your comments.

Vinay Nandicoori